# Synucleinopathy alters nanoscale organization and diffusion in the brain extracellular space through hyaluronan remodeling

Federico N. Soria [1,2,3], Chiara Paviolo[4,5], Evelyne Doudnikoff[1,2], Marie-Laure Arotcarena [1,2],
Antony Lee [4,5], Noémie Danné [4,5], Amit Kumar Mandal[4,5], Philippe Gosset [1,2], Benjamin Dehay [1,2],
Laurent Groc[6,7], Laurent Cognet [4,5✉] & Erwan Bezard [1,2✉]

In recent years, exploration of the brain extracellular space (ECS) has made remarkable progress, including nanoscopic characterizations. However, whether ECS precise conformation is altered during brain pathology remains unknown. Here we study the nanoscale organization of pathological ECS in adult mice under degenerative conditions. Using electron microscopy in cryofixed tissue and single nanotube tracking in live brain slices combined with super-resolution imaging analysis, we find enlarged ECS dimensions and increased nanoscale diffusion after α-synuclein-induced neurodegeneration. These animals display a degraded hyaluronan matrix in areas close to reactive microglia. Furthermore, experimental hyaluronan depletion in vivo reduces dopaminergic cell loss and α-synuclein load, induces microgliosis and increases ECS diffusivity, highlighting hyaluronan as diffusional barrier and local tissue organizer. These findings demonstrate the interplay of ECS, extracellular matrix and glia in pathology, unraveling ECS features relevant for the α-synuclein propagation hypothesis and suggesting matrix manipulation as a disease-modifying strategy.

[1] Université de Bordeaux, Institut des Maladies Neurodégénératives, UMR 5293, 33076 Bordeaux, France. [2] Centre National de la Recherche Scientifique, IMN, UMR 5293, 33076 Bordeaux, France. [3] Achucarro Basque Center for Neuroscience, Universidad del País Vasco (UPV/EHU), 48940 Leioa, Spain. [4] Université de Bordeaux, Laboratoire Photonique Numérique et Nanosciences, UMR 5298, 33400 Talence, France. [5] Institut d'Optique & Centre National de la Recherche Scientifique, LP2N, UMR 5298, 33400 Talence, France. [6] Université de Bordeaux, Interdisciplinary Institute for Neuroscience, UMR 5297, 33076 Bordeaux, France. [7] Centre National de la Recherche Scientifique, IINS, UMR 5297, 33076 Bordeaux, France. ✉email: laurent.cognet@u-bordeaux.fr; erwan.bezard@u-bordeaux.fr

The brain extracellular microenvironment is the active milieu surrounding cellular membranes in the central nervous system (CNS). It is composed by a dynamic compartment, the extracellular space (ECS), which contains the interstitial fluid and a plastic scaffold known as the extracellular matrix (ECM). The ECS provides a reservoir of ions to support membrane and action potentials, playing a fundamental role in the diffusion of chemical signals and intercellular communication[1–3]. Additionally, the ECS is thought to be implicated in the large-scale advection of molecules by determining the bulk flow of interstitial fluid[4,5], possibly through convective or local flow[6]. With a width down to the nanometer range, the ECS has remained a black box in neuroscience until very recently[7], and imaging the adult brain ECS is still a challenging endeavor.

The main structural component of the brain ECS is the ECM, a dense network of macromolecules secreted by glia and neurons[8]. Unlike the ECM from connective tissue, where collagen is the main unit, the neural interstitial matrix mainly consists of long chains of the glycan polymer hyaluronan[9]. Other ECM components bind to large hyaluronan molecules forming a self-assembled matrix that functions as cell-attaching scaffold and maintains the patency of the ECS, influencing its organization at a multilevel scale[7]. Interestingly, hyaluronan has also manifold signaling properties depending on its size[10]. High-molecular weight-hyaluronan (HMW-HA) elicits anti-inflammatory and anti-proliferative responses, whereas short hyaluronan fragments activate pro-inflammatory pathways in specific cells such as microglia[11]. Although the function of ECM during CNS development and plasticity is well documented, major questions remain about its involvement in the pathophysiology of the mature brain[12].

The implications of this complex extracellular system on the etiology and progression of brain disorders remain largely underexplored. While limited information on ECM dynamics exists in selected neuropathologies such as multiple sclerosis and stroke, data on Parkinson's disease (PD) or other proteinopathies is scarce[12,13]. Furthermore, as we enter the realm of super-resolved ECS exploration[14], our data on the pathological brain ECS is incomplete. In healthy tissue, ECS nanoscale topology has been described both in cryofixed[15–17] and live samples[18]. Moreover, we have recently reported the ECS local nanoscale organization and diffusivity on young[2] and adult live brain[19] using single-walled carbon nanotubes (SWCNTs) as super-resolution imaging probes. Whether these parameters change during disease states and whether an interplay exists between the ECS and the pathological process is still unknown.

Here we report on the dynamics and organization of the brain extracellular microenvironment in a PD pathological context, using a combination of state-of-the-art imaging approaches and an established in vivo model of α-synuclein (α-syn)-induced dopaminergic neurodegeneration[20,21], a proxy of synucleinopathy leading to PD. We describe profound alterations in ECS morphological and diffusional parameters in the *substantia nigra* of parkinsonian mice at nanometer resolution. We further characterize the pathological ECS by exploring the status of the hyaluronan network in vivo, its relation with reactive microglia and inflammation, and dopaminergic cell loss and ECS parameters after acute and chronic hyaluronan depletion. Our results reveal an interplay between ECS, ECM, and neurodegeneration, shedding light upon a neglected compartment for the diffusion of aggregated α-synuclein seeds. This integrative study explores the pathological extracellular microenvironment as a whole in adult brain tissue, and paves the way to explore the ECS in other models of proteinopathies.

## Results

**α-syn-induced neuronal loss enlarges the extracellular space.** To investigate the ECS in a context of neurodegeneration, we employed a unique paradigm of α-syn-induced dopaminergic neuronal loss by unilateral inoculation of Lewy body (LB) fractions derived from PD patients into the *substantia nigra* (SN) of adult mice[20,21]. Control animals received fractions containing only soluble α-syn (noLB), while the LB fractions contained α-syn seeds in its aggregated form. Stereological cell counts of tyrosine hydroxylase (TH) immunostaining confirmed 47% nigral dopaminergic degeneration (ipsi/contra TH-positive cell ratio: $0.96 \pm 0.05$ noLB, $0.53 \pm 0.05$ LB) 4 months after LB inoculation (Fig. 1a, b and Supplementary Fig. 1), a similar rate to the histopathological features of this model, that have been extensively described elsewhere[20,21].

Electron microscopy (EM) has been historically used to study the topology of brain tissue at sub-micron resolution. However, ultrastructural analysis of the ECS has proven difficult to implement due to dehydration during fixation and sample processing, which shrinks the ECS and underestimates its size. While isotonic membrane-impermeant buffers have been used to overcome this artifact with relative success[22,23], cryofixation studies have shown effective ECS preservation in cerebellum[15] and mouse neocortex[16,17]. We, therefore, used cryofixation of brain tissue rapidly removed from LB-inoculated mice to reveal the ECS in the SN. High-pressure cryofixation prevents formation of ice crystals and membrane rupture and freeze-substitution resin embedding retains molecules in their hydrated positions, effectively preserving the extracellular space when compared to classical aldehyde fixation (Fig. 1c). The result is a more accurate representation of the ECS, where key elements of the neuropil retain their physiological dimensions (Supplementary Fig. 2a). This preparation was able to preserve ECS heterogeneity as we could discern, for instance, compact intermembrane gaps at the synaptic cleft as well as large perisynaptic spaces surrounding them (Supplementary Fig. 2b, c). Details were sufficiently clear to delineate the ECS and resolve extracellular gaps of 20 nm width (Supplementary Fig. 2d).

With this tool in hand, we quantitatively analyzed the ECS dimensions in the SN of control and LB-inoculated mice. After the manual segmentation of EM ultramicrographs, we calculated the relative ECS area (Fig. 1d). This estimation of ECS volume fraction (defined as $\alpha$ in previous studies) showed variations in the SN between 15% and 24%, in accordance with previously reported $\alpha$ values for other brain regions in mice[1]. There was a clear tendency for larger ECS volume fractions in the SN of LB-inoculated mice ($16.8 \pm 1.1\%$ noLB, $20.0 \pm 1.0\%$ LB; Fig. 1e) although the difference regarding control nearly achieved statistical significance ($P = 0.064$). Since volume fraction is a global parameter that takes into account the region sampled as a whole, to take advantage of the nanometric resolution achieved by EM we next explored local geometric variations by analyzing individual 2D compartments in binary images of the ECS (Fig. 1d). Skeletonization revealed heterogeneous lengths, detecting significantly longer ECS compartments in parkinsonian mice when compared to control ($0.52 \pm 0.05$ μm noLB, $0.70 \pm 0.12$ μm LB; Fig. 1f). Analysis of 2D-local thickness identified larger ECS widths in the SN of LB-inoculated mice when compared to control ($112 \pm 7$ nm noLB, $140 \pm 6$ nm LB; Fig. 1g). Mapping local ECS widths yielded diverse geometries, where large pools with diameters around 200 nm are connected by narrow channels of less than 100 nm width (Fig. 1h). The distribution of ECS widths was positively skewed, with ~50% values under 100 nm (Fig. 1i). Differences were greater in the larger dimensions (>200 nm), as the SN of LB-inoculated mice displayed twice the number of large pools than control animals (11.83% noLB, 20.61% LB; Fig. 1j).

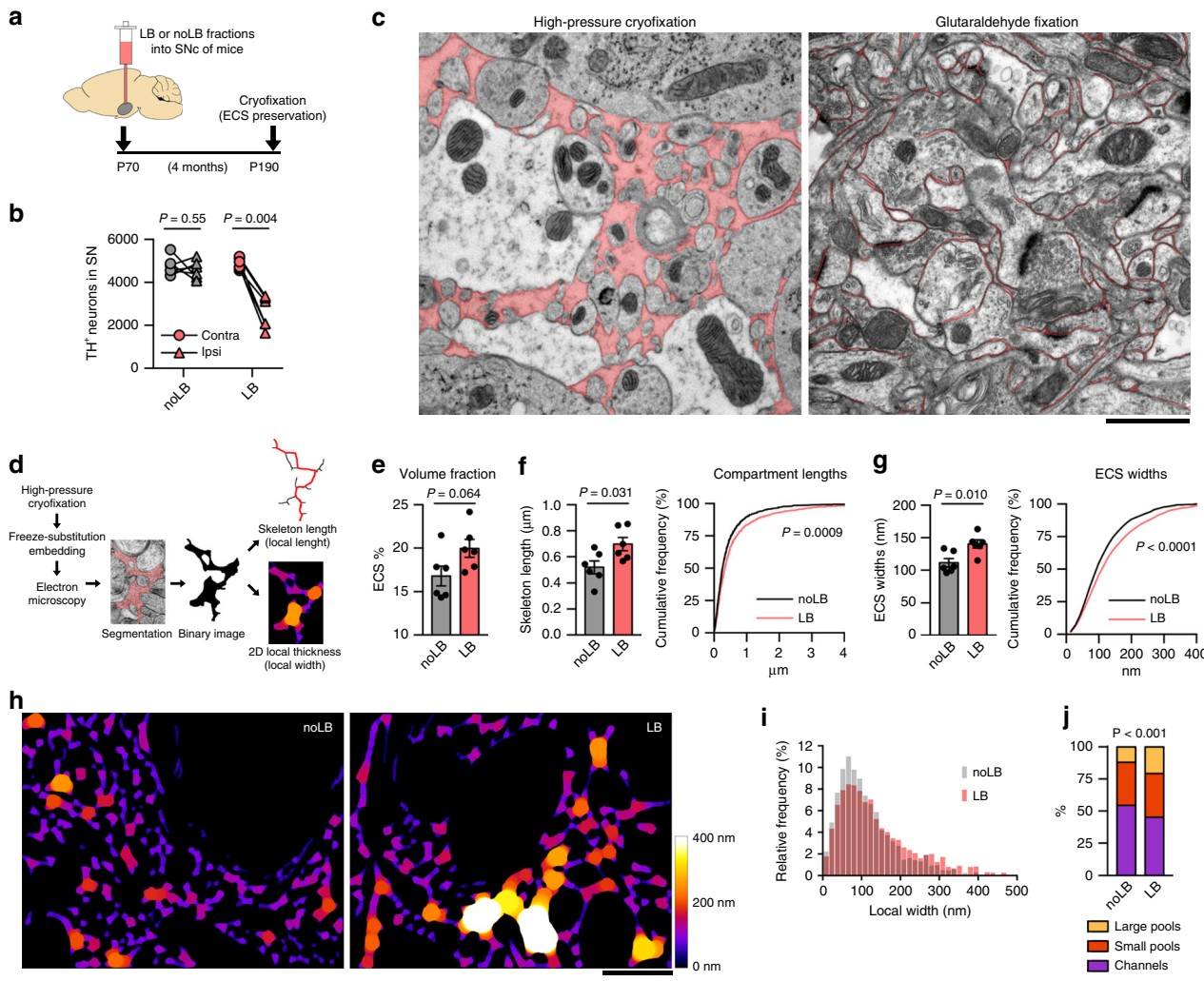

**Fig. 1 α-syn-induced neurodegeneration enlarges the extracellular space locally. a** Experimental timeline of the mouse model of Lewy body (LB)-induced neurodegeneration. **b** TH stereological cell counts confirmed 40% dopaminergic cell loss typically observed in the SN of LB-inoculated mice (paired two-tailed Student's $t$ test, $n = 6$ mice). **c** Ultramicrographs of cryofixed (left) vs chemically fixed (right) brain tissue (ECS is pseudo-colored). Classic glutaraldehyde fixation artefactually reduces ECS size, whereas high-pressure cryofixation represents more accurately ECS dimensions. Scale bar, 1 μm. **d** Schematic diagram of cryofixation, freeze-substitution, EM and image analysis protocol. **e** ECS volume fraction in the SN of LB and noLB-mice (two-tailed Student's $t$ test, $n = 6$ SNs). **f**, **g** Mean and cumulative distribution of ECS compartment lengths (**f**) and local widths (**g**) in the SN after LB-induced neurodegeneration (Two-tailed Student's $t$ test and Kolmogorov–Smirnov test, $n = 6$ SNs). **h** ECS local width maps from Ctrl and LB-inoculated mice display the characteristic channels-and-pools configuration of the ECS, revealing increased dimensions in the degenerated SN. Scale bar, 1 μm. **i** Distribution of ECS local widths from all maps generated in the study. **j** Proportions of ECS compartments categorized by width (Large pools, >200 nm; small pools, 100–200 nm; channels, <100 nm) expressed as % of ECS (Chi-square test) showing that LB-induced neurodegeneration affects how the ECS is spatially distributed. Error bars represent mean ± SEM. Source data are provided as a Source data file. See also Supplementary Figs. 1 and 2.

Thus, neurodegeneration would affect how the ECS volume fraction is spatially distributed, rather than the volume fraction per se.

**Synucleinopathy increases extracellular space diffusivity.** Molecular mobility through the ECS is key to important physiological events, such as signal transmission or ion concentration[24]. Hence, determining live ECS local diffusive environment is as essential as knowing its dimension. We have recently used single-molecule tracking of fluorescent SWCNTs to quantitatively visualize nanoscale dimensions and diffusivity in the adult mouse ECS by near-infrared (NIR) imaging[19]. Here, we used acute slices of LB-inoculated mice (i.e., 4 months post-inoculation) 1 h after in vivo injection of SWCNTs to explore local ECS parameters in a neurodegenerative context (Fig. 2a, b). As non-internalizable ECS fluorescent probes of large aspect-ratio[2], Brownian motion of

SWCNTs is effectively constrained by brain ECS geometry and composition with respect to free medium (Supplementary Movie 1). SWCNTs exceptional photostability enables long-term acquisitions with negligible decrease in fluorescence (Supplementary Movie 2). However, since they rarely remain confined to a specific area, individual acquisitions of 5–10 min were performed. Single-molecule localization allowed for sub-wavelength (~50 nm) tracking of SWCNTs and provided super-resolution analysis of ECS dimensions. In addition, iterative analysis of finite time-windows rendered instantaneous relative diffusivity ($D_{inst}/D_{ref}$) maps of the nigral ECS with nanoscale precision (Supplementary Fig. 3).

Local instantaneous diffusivity maps revealed a diverse ECS in both healthy and degenerative SN, where diffusional patterns change drastically within few nanometers (Fig. 2c and Supplementary Fig. 4). SWCNT tracks revealed two different

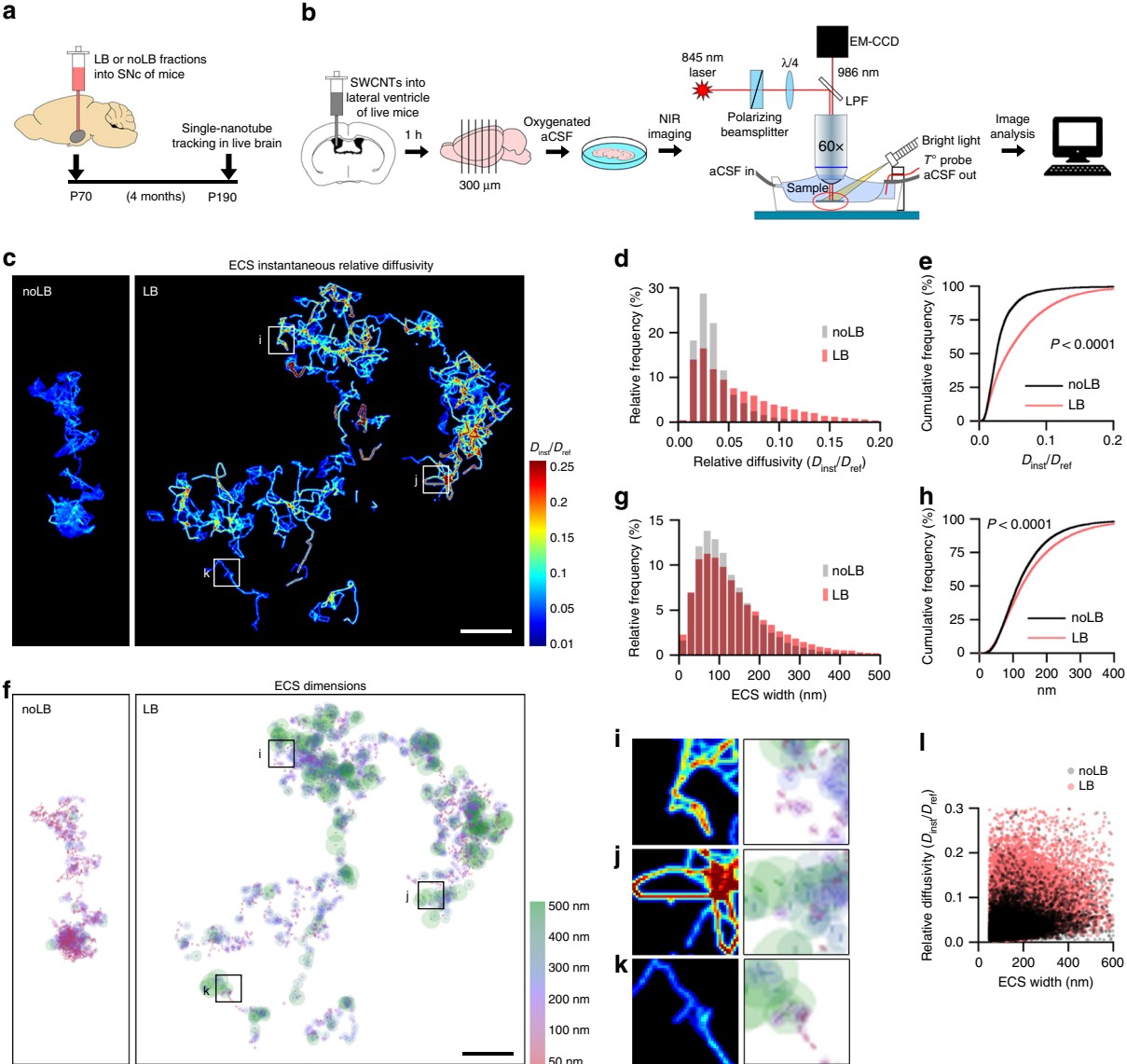

**Fig. 2 Single-nanotube tracking reveals increased ECS nanoscale diffusivity and width after synucleinopathy. a** Experimental timeline. **b** Schematic diagram of SWCNT imaging and tracking setup. Fluorescent SWCNTs are administered in vivo into the lateral ventricle of adult mice and allowed to diffuse into the brain for 1 h. Acute brain slices are imaged at 37 °C using a NIR fluorescence microscope equipped with an EM-CCD camera. **c** ECS instantaneous relative diffusivity maps reveal a heterogeneous ECS in the SN, with increased nanoscale diffusion after LB-induced neurodegeneration. Scale bar, 2 μm. **d**, **e** Frequency (**d**) and cumulative (**e**) distributions of local relative diffusivity values ($D_{inst}/D_{ref}$) in the nigral ECS indicate augmented nanoscale diffusion in the parkinsonian SN (Kolmogorov–Smirnov test, $n = 4$ mice). **f** Maps of local ECS dimensions in the SN of noLB and LB-inoculated mice, corroborating in live brain the channels-and-pools configuration observed in cryofixed tissue. Scale bar, 2 μm. **g**, **h** Frequency (**g**) and cumulative (**h**) distributions of local ECS dimensions show increased ECS width in LB-inoculated mice (Kolmogorov–Smirnov test, $n = 4$ mice). **i**–**k** Insets from (**c**, **f**). Increased diffusion is not always correlated to enlarged dimensions, suggesting that inhomogeneous diffusion is not only dependent on environmental geometry. **l** Scatter plot of all trajectories showing weak correlation between both parameters. Source data are provided as a Source data file. See also Supplementary Figs. 3 and 4.

configurations: long explorations spanning several μm, with ECS channels connected by pools in a similar fashion to EM images (Supplementary Fig. 4a), and spatially restricted compartments where SWCNTs move within the same domain (Supplementary Fig. 4b). Quantitative analysis detected a highly constrained ECS in the control SN, where SWCNT movement was hindered more than 10-fold respect to free medium ($D_{inst}/D_{ref}$ median = 0.026, IQR = 0.017-0.038; Fig. 2d, e). On the contrary, in LB-inoculated mice we observed increased local diffusivity respect to control, with a significant proportion of values falling within the 0.1–0.2 range (median = 0.042, IQR = 0.021−0.080; Fig. 2d, e). Furthermore, in LB-exposed animals, SWCNTs exploration covered a significantly broader ECS area within the same time-frame (6.2 ±

1.4 μm² noLB, 16.2 ± 4.0 μm² LB; Supplementary Fig. 4c and Supplementary Movie 3). These results indicate increased ECS nanoscale diffusion after LB-induced neurodegeneration.

In a similar fashion to traditional ECS diffusional studies[25], single SWCNT trajectories can reveal the nanoscale geometry of the compartment explored by applying a super-resolution microscopy methodology. Dimensional maps in the live brain (Fig. 2f) confirmed the nanoscale channels-and-pools ECS configuration, whereas quantitative analysis corroborated EM results, revealing increased local ECS widths after LB-induced neurodegeneration (median = 114 nm, IQR = 76–170 nm noLB, median = 119 nm, IQR = 75–187 nm LB; Fig. 2g, h). Interestingly, we found weak correlation between ECS dimensions and

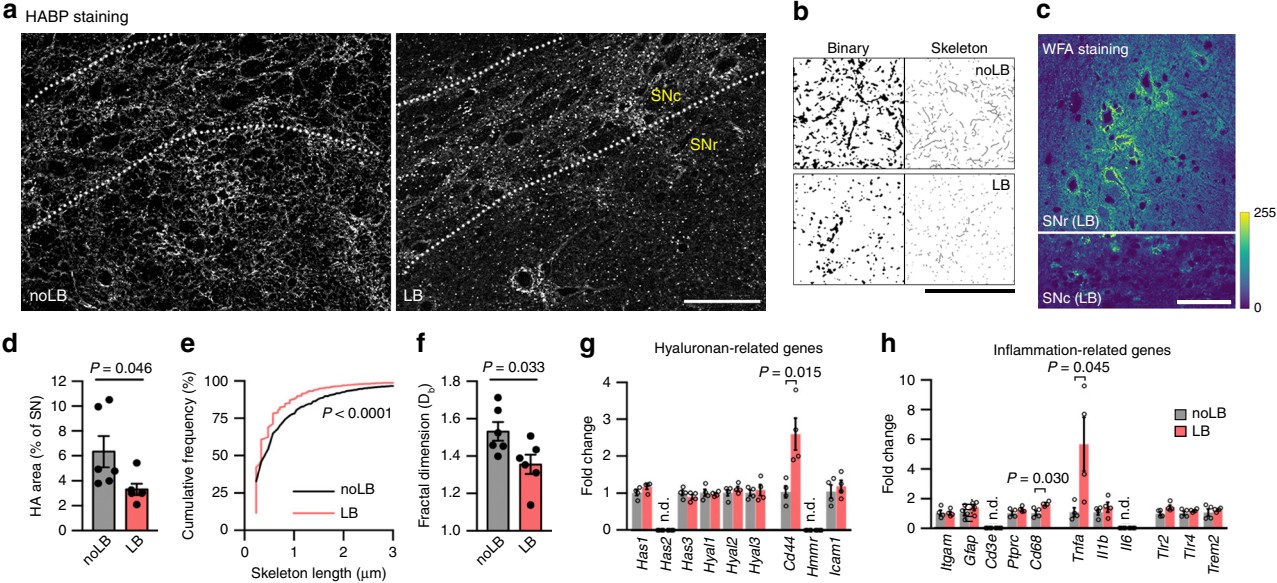

**Fig. 3 The hyaluronan matrix is degraded after synucleinopathy. a** Confocal micrographs from the SN of noLB and LB-inoculated mice, showing a widespread disruption of the matrix in the degenerated SN and fewer cable-like HA structures. Scale bar, 50 μm. **b** Binary and skeletonized images of the HA network used to quantify HA matrix disruption. Note the dotted, degraded pattern of HA in the degenerated SN. Scale bar, 50 μm. **c** WFA staining in the SN of LB-inoculated mice reveals no disruption of CSPGs. Scale bar, 50 μm. Color bar represents fluorescence intensity. **d** HA levels are decreased in the SN of LB-inoculated mice (two-tailed Student's *t* test, *n* = 6 mice). **e** Cumulative distributions of HA skeleton lengths indicate shorter cable-like structures after LB-induced neurodegeneration (Kolmogorov–Smirnov test, *n* = 6 mice). **f** Fractal dimension analysis of the HA network shows disrupted matrix interconnectivity in the parkinsonian SN (two-tailed Student's *t* test, *n* = 6 mice,). **g, h** mRNA expression analysis of HA-related genes (**g**) and inflammation-related genes (**h**) in ventral midbrain homogenates of LB and noLB-inoculated mice (two-tailed Student's *t* test, *n* = 4 mice). Error bars represent mean ± SEM. Source data are provided as a Source data file. See also Supplementary Fig. 5.

diffusivity (Pearson's *r* = 0.195 noLB, 0.353 LB; Fig. 2i–l), suggesting that spatial constriction is not necessarily the central determinant of ECS diffusion inhomogeneities at the nanoscale.

**Synucleinopathy induces degradation of the hyaluronan matrix.** The main diffusional barrier in the brain ECS is the neural interstitial matrix[1,24]. As its main scaffolding molecule, changes in hyaluronan (HA) content can dramatically alter other ECM components[26] and consequently ECS parameters[2,27]. We, therefore, analyzed the status of the HA network in the mouse SN after LB-induced neurodegeneration by confocal microscopy. Hyaluronic acid-binding protein (HABP) staining in the murine SN was more intense than in surrounding regions, but the staining pattern differed between subareas (Supplementary Fig. 5a). The *substantia nigra pars compacta* (SNc), which harbors packed dopaminergic neurons projecting mainly to the striatum, presented a dense interstitial HA matrix (Supplementary Fig. 5b). In the *substantia nigra pars reticulata* (SNr), which contains dendrites radiating from the SNc and scattered GABAergic neurons, HA staining was more disperse but especially intense in the perineuronal nets (PNNs; Supplementary Fig. 5b).

This precise organization was substantially altered after LB-induced neurodegeneration. The interstitial matrix appeared more diffuse, and the HA cable-like structures often described in literature[28] were limited to dispersed dots (Fig. 3a, b). Despite neurodegeneration and HA disruption, chondroitin-sulfate proteoglycans (CSPGs) appeared unaffected as observed by the archetypical staining of PNNs by Wisteria Floribunda Agglutinin (WFA; Fig. 3c), as reported by others[29–31]. Quantification detected decreased HA immunostaining in the SN of LB-inoculated mice compared to control (3.3 ± 0.4% vs 6.3 ± 1.2%; Fig. 3d). Morphological analysis revealed shorter HA cables in LB-inoculated mice (median = 0.48 μm noLB, 0.34 μm LB; Fig. 3e). Despite HA being a linear polysaccharide, it appears as

an interconnected network thanks to cross-linking by hyalectans and tenascins, forming supramolecular complexes of varying stability[32]. Measuring HA cross-linking provide, thus, an estimation of matrix interconnectivity and organization. We, therefore, quantified HA network complexity by fractal analysis ($D_b$)[33] and detected lower values in LB-inoculated mice (1.53 ± 0.04% noLB, 1.35 ± 0.05% LB; Fig. 3f), indicating decreased HA network complexity in the degenerated SN.

To investigate the basis for HA disruption, we next analyzed mRNA expression of genes related to HA synthesis/catabolism in ventral midbrain homogenates from control and LB-inoculated mice. We detected no differences in mRNA expression of HA synthases (*Has*) or hyaluronidases (*Hyal*). Since HA catabolism also depends on receptor-mediated endocytosis[34], we examined mRNA levels of HA receptors. We found increased expression of CD44 (2.6 ± 0.4-fold; Fig. 3g), the main HA receptor, expressed mostly in glia and associated with inflammation in the CNS[10,35]. Additionally, we tested several glia and inflammation-related genes (Fig. 3h), detecting a significant increase in mRNA levels of microglia/macrophage activation marker CD68 (1.6 ± 0.1-fold change) and pro-inflammatory cytokine TNF-α (5.7 ± 1.8-fold change), indicate ongoing inflammation in the SN after LB-induced neurodegeneration.

**Reactive microglia engulf hyaluronan after synucleinopathy.** Whereas reactive microglia has been described in several PD animal models, experimental data on non-transgenic α-synuclein-based PD models is scarce[36]. We, therefore, characterized the microglial phenotype after LB-induced neurodegeneration by immunohistochemistry. In control mice, Iba1 immunostaining was restricted mostly to the SNr, with few microglial somas in the SNc (Fig. 4a, b), as described in literature[37]. This pattern was substantially altered in LB-inoculated mice, with more than twofold microglial cells (132 ± 14 noLB, 267 ± 29 LB; Fig. 4c) and

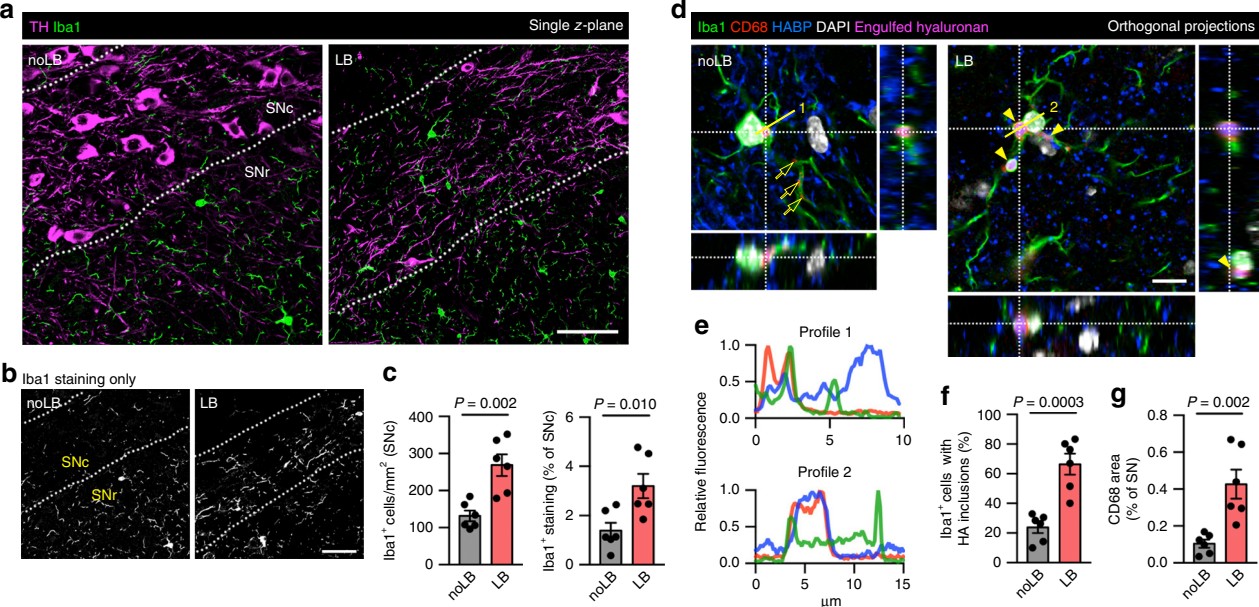

**Fig. 4 Reactive microglia populate the SNc and phagocytes hyaluronan after α-syn-induced neurodegeneration. a, b** Confocal images showing distribution of microglia (Iba1) in the SN of mice. In control animals, microglia populate mostly the SNr and is scarce in the SNc, packed with TH-positive neurons. This pattern is disrupted in LB-inoculated mice. Scale bars, 50 μm. **c** Quantification of Iba1-positive cells and Iba1 area in the SNc show microgliosis in this region after LB-induced neurodegeneration (two-tailed Student's t test, n = 6 mice). **d** Orthogonal projections showing HA phagocytosis by microglia in the murine SN. Engulfed HA co-localizes with lysosomal marker CD68 (arrowheads), although some CD68 dots are negative for HABP (arrows). Microglia from LB-inoculated mice display more HA inclusions, as seen in this micrograph where up to 4 pouches of engulfed HA can be observed. Scale bar, 10 μm. **e** Fluorescence intensity profiles from (**d**). **f** Quantification of microglia with HA inclusions in the SN of LB and noLB-mice (two-tailed Student's t test, n = 6 mice). **g** Quantification of CD68 levels indicate microglial activation in the SN of LB-inoculated mice (two-tailed Student's t test, n = 6 mice). Error bars represent mean ± SEM. Source data are provided as a Source data file. See also Supplementary Fig. 6.

Iba1 staining (1.4 ± 0.3% noLB, 3.2 ± 0.5% LB; Fig. 4c) in the SNc. Microgliosis was evident in SNc areas depleted of TH-positive cells, but also in regions where dopaminergic cell loss was milder (Supplementary Fig. 6a, b).

We next examined the activation status and phagocytic activity of microglia in a context of nigral matrix degradation by multiple immunostaining. We detected HA particles engulfed by microglia in both noLB and LB-inoculated mice (Fig. 4d). HABP staining inside Iba1-positive cells co-localized with lysosomal marker CD68 (Fig. 4d, e). Although not all CD68-positive pouches were positive for HA, both HA-positive and HA-negative pouches were found in the same cell (Supplementary Fig. 6c, d). Despite these results suggesting a physiological role of microglia in HA turnover, the number of microglial cells with HA inclusions was significantly increased after LB-induced neurodegeneration (23.5 ± 3.8% noLB, 66.1 ± 7.2% LB; Fig. 4f), often displaying more than one HA-positive pouch per cell (Fig. 4d, arrowheads). Finally, CD68 staining was significantly higher in LB-inoculated mice (0.43 ± 0.07% vs. 0.10 ± 0.02%; Fig. 4g), confirming qPCR data and indicating microglial activation in the lesioned SN. Taken together, these results suggest an involvement of reactive microglia in HA matrix degradation after LB-induced dopaminergic cell loss.

**Hyaluronan breakdown is neuroprotective in LB-injected mice.** Having described the consequences of neurodegeneration on the brain extracellular microenvironment, we next enquired whether modification of ECS/ECM would alter the pathological process. Degradation of the HA matrix by intracerebral injection of hyaluronidase (Hyase) has proven a useful tool to explore ECM modification in vivo[30,38], and we have previously reported its effect by observing increased ECS nanoscale diffusion in the rodent brain[2]. We, therefore, inoculated LB fractions concomitant

with Hyase (or PBS as control) in the murine SN (Fig. 5a). Hyase did not alter the aggregation status of LB fractions, as observed by unchanged thioflavin T fluorescence profiles (Supplementary Fig. 7a). Four months post-inoculation, LB + PBS group displayed 40% neurodegeneration (Fig. 5b–d), similar to the classical LB-inoculation protocol. In contrast, stereological cell counts revealed only 20% dopaminergic cell loss in LB + Hyase-inoculated mice (2736 ± 257 cells LB + PBS, 3957 ± 318 LB + Hyase; Fig. 5c, d). We noticed fewer Proteinase K (PK)-resistant α-syn aggregates in LB + Hyase-treated mice (Supplementary Fig. 7b), although the low number of aggregates rendered any quantification futile. To evaluate whether the neuroprotective effect of Hyase was exclusively related to HA, we co-inoculated LB with chondroitinase (ChABC) to degrade CSPGs, another ECM component associated to structural HA. ChABC injection in vivo effectively removed PNNs in the SN (Supplementary Fig. 8), but did not alter the typical 0.6 ipsi/contra ratio triggered by LB (Fig. 5c, d), indicating that Hyase effect is connected to HA degradation rather than to ECM disorganization. Co-injection of LB with high-molecular weight HA (HMW-HA) also failed to reduce LB-induced neurodegeneration (Fig. 5c, d). However, concomitant injection of LB with low-molecular weight HA (LMW-HA) ameliorated dopaminergic cell loss to a similar extent than Hyase treatment (3560 ± 144 cells; Fig. 5c, d), suggesting that HA breakdown into small fragments is responsible for the neuroprotective effect, instead of merely a modification of HA content.

Hyase effect in the rodent brain lasts only a few days, due to the high replenishing ratio of HA in vivo which reappears as isolated patches 3–5 days after injection[38]. Furthermore, proteopathic seeding from exogenous α-syn typically occurs within 24 h after inoculation of LB fractions[20]. We, therefore, investigated the early events (72 h) after LB + Hyase inoculation (Fig. 5e). Human

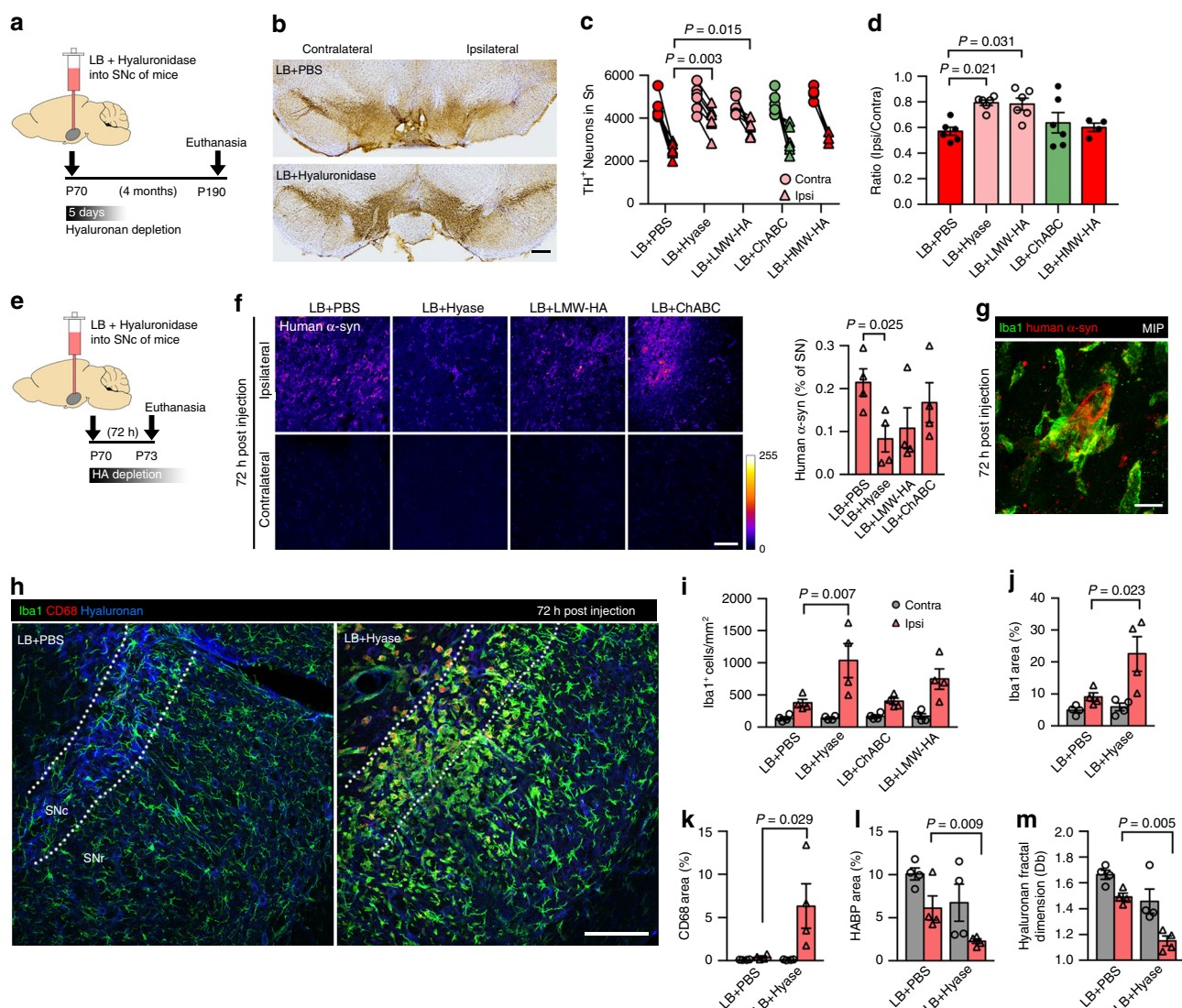

**Fig. 5 Hyaluronan fragmentation ameliorates LB-induced neurodegeneration by reducing toxic-α-syn load. a** Experimental timeline for (**b–d**). Hyaluronidase (Hyase) effect lasts only a few days post-injection, since HA is rapidly replenished in vivo. **b** TH staining shows partial neuroprotection in the ipsilateral SN 4 months after LB + Hyase inoculation. Scale bar, 200 μm. **c, d** Stereology revealed a reduction (from 40% to 20%) in dopaminergic cell loss when LB were co-injected with Hyase or low-molecular weight HA (LMW-HA), but not with chondroitinase ABC (ChABC), suggesting an effect mediated by HA breakdown. Graphs show TH-positive cell count (**c**) and ipsilateral/contralateral TH-positive cell ratio (**d**) (One-way ANOVA with Tukey's post-hoc test, n = 4–6 mice). **e** Experimental timeline for (**f–m**). Animals were euthanized at the peak of Hyase effect. **f** Confocal micrographs and quantification of human α-syn staining in the SN 72 h after LB + Hyase injection show decreased exogenous α-syn load. Scale bar, 50 μm. Color bar represents fluorescence intensity. **g** Microglial cells surround a human α-syn-positive cell. Scale bar, 10 μm. **h** Substantial microglial activation in LB + Hyase-treated SN shortly after injection, suggesting an early effect of HA cleavage. Scale bar, 100 μm. **i–l** Quantification of Iba1-positive cells, Iba1, CD68 and HABP levels 72 h after LB inoculation co-injected with various ECM-modification molecules (two-way ANOVA with Holm–Sidak's post-hoc test, n = 4 mice). **m** Fractal analysis of HABP staining confirms HA matrix disruption by Hyase (two-way ANOVA with Holm–Sidak's post-hoc test, n = 4 mice). Data expressed as mean ± SEM. Source data are provided as a Source data file. See also Supplementary Figs. 7 and 8.

α-syn staining was significantly lower in the SN after LB + Hyase injection in comparison to LB + PBS (0.08 ± 0.03% vs. 0.21 ± 0.03%; Fig. 5f), indicating reduced patient-derived α-syn load in Hyase-treated animals. This decrease in α-syn inoculum was partially reproduced by LMW-HA co-injection, but not by ChABC treatment (Fig. 5f). Human α-syn staining in the contralateral side was absent (Fig. 5f), demonstrating antibody specificity. We did not find human α-syn engulfed by microglia, although we did observe human α-syn-positive cells surrounded by multiple Iba1-positive somas (Fig. 5g), suggesting microglia can somehow sense damaged or infected cells, a hypothesis recently demonstrated in experimental stroke[39]. Moreover,

Iba1 staining showed early microglial invasion into the LB + PBS-treated SNc, as previously seen for 4-month animals (Fig. 5h), suggesting that microglial activation takes place shortly after LB inoculation. Interestingly, LB + Hyase-treated animals displayed several-fold higher microglial activation in the ipsilateral SN compared to LB + PBS, as indicated by Iba1-positive cell count (1036 ± 261 cells vs. 377 ± 56; Fig. 5i), Iba1 staining (22.6 ± 5.4% vs. 9.1 ± 1.3%; Fig. 5j) and CD68 staining (6.3 ± 2.6% vs. 0.4 ± 0.1%; Fig. 5k). Microgliosis was partially reproduced in the LB + LMW-HA group, but not in the LB + ChABC group (Fig. 5i), suggesting that HA small fragments might be driving neuroinflammation. In fact, HABP labeling

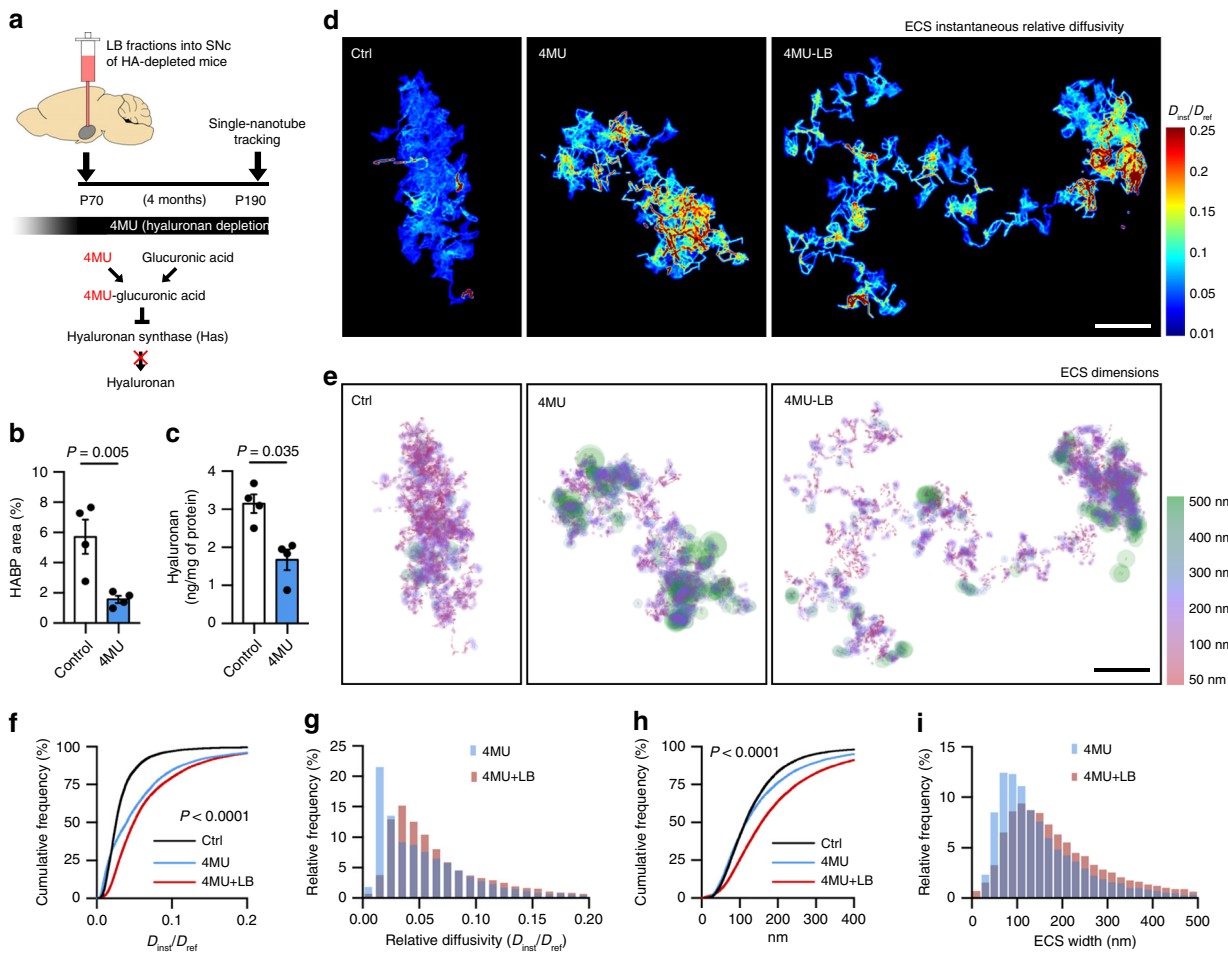

**Fig. 6 Hyaluronan depletion increases ECS nanoscale diffusivity more profoundly than ECS width. a** Experimental timeline. 4-methylumbelliferone (4MU) inhibits HA synthesis in vivo. 4MU-based diet started 4 weeks prior to LB-inoculation to deplete HA chronically before the onset of pathology. **b**, **c** Quantification of HA staining in SN of mice (**b**) or HA content in mouse plasma (**c**) after 1 month of 4MU or normal diet (two-tailed Student's $t$ test, $n = 4$ mice). **d**, **e** ECS relative diffusivity (**d**) and spatial (**e**) maps of the nigral ECS in control mice and mice with experimental matrix disruption (4MU) and experimental + pathological degradation of the HA matrix (4MU-LB). Scale bars, 2 μm. **f** Cumulative distributions of ECS relative diffusivities show increased nanoscale diffusion in the SN of 4MU-treated mice and 4MU-mice after LB-induced neurodegeneration (Kolmogorov–Smirnov test of 4MU vs 4MU-LB, $n = 4$ mice). **g** Frequency distributions of nigral ECS $D_{inst}/D_{ref}$ obtained from 4MU and 4MU-LB-treated mice. **h** Cumulative distributions of ECS dimensions show increased ECS width in the SN of 4MU-mice, an effect augmented in 4MU-treated mice injected with LBs (Kolmogorov–Smirnov test of 4MU vs 4MU-LB, $n = 4$ mice), suggesting that neurodegeneration impacts ECS width, while HA depletion influences mostly ECS diffusivity. **i** Frequency distributions of ECS widths show an increased ECS enlargement in the degenerated (LB) and HA-depleted (4MU) SN, with most dimensions exceeding 100 nm. Error bars represent mean ± SEM. Source data are provided as a Source data file. See also Supplementary Figs. 9 and 10.

confirmed HA depletion ($10.1 \pm 0.7\%$ LB + PBS$_{contra}$, $2.3 \pm 0.3\%$ LB + Hyase$_{ipsi}$; Fig. 5l) whereas fractal analysis detected heavy HA network disruption 72 h after Hyase injection (Fig. 5m). Taken together, these results indicate that acute HA depletion, most certainly via HA fragmentation, exacerbates the early (72 h) microglial activation observed after LB inoculation and reduces proteopathic seeds load. These events ultimately result in long-term neuroprotection (i.e., 4 months).

**Hyaluronan depletion affects mostly ECS diffusivity.** We next explored the effect of chronic HA modification on ECS parameters and on the pathological process. Long-term HA depletion was achieved by chronic treatment with 4-methylumbelliferone (4MU, Fig. 6a), a *Has* inhibitor extensively used in vivo[40]. HA depletion was confirmed in blood samples by ELISA and by immunohistochemistry in different brain regions (Fig. 6b, c and Supplementary Fig. 9). First, we checked whether chronic HA depletion alters ECS local diffusion in the SN of LB-inoculated mice by SWCNT tracking in acute slices. We observed increased relative

diffusivity in 4MU-treated animals with respect to normal diet, an increase slightly exacerbated after neurodegeneration (median = 0.025 Ctrl, 0.040 4MU, 0.049 4MU-LB; Fig. 6d, f, g and Supplementary Movie 4). ECS widths were also increased in both 4MU and 4MU + LB groups. However, ECS enlargement was much more pronounced with the additional presence of LB (median = 135 nm Ctrl, 154 nm 4MU, 195 nm 4MU-LB; Fig. 6e, h, i), suggesting that neurodegeneration has more impact on ECS morphology (widths), while ECS diffusivity depends mostly on HA fluctuation (Fig. 6f vs. 6h). Finally, we examined nigral neurodegeneration at 4 months post-LB inoculation, and we detected a non-significant trend for neuroprotection in the 4MU-treated animals with respect to normal diet ($35.9 \pm 7.6\%$ 4MU-LB, $45.1 \pm 6.8\%$ Ctrl-LB; Supplementary Fig. 10a, b), indicating that chronic HA depletion has milder effects on dopaminergic cell loss than acute depletion with Hyase, probably due to compensatory mechanisms. Widespread microgliosis was observed after 1 month of 4MU-dependent HA depletion (Supplementary Fig. 10c), as evidenced by increased Iba1 staining and Iba1-positive somas in

both ipsi and contralateral sides 72hs post LB-injection (Supplementary Fig. 10d). Chronic neuroinflammation, rather than acute, may complicate even further the neuropathological outcome, and hence it might explain the lack of neuroprotection in the 4MU model. Taken together, these results indicate that experimental chronic HA depletion increases ECS width and in particularly diffusivity, in a similar fashion to pathological (LB-induced) HA degradation. This confirms that ECS local parameters, and especially nanoscale diffusion, are inversely related to HA.

## Discussion

Unravelling the ECS structure and function represents a technological challenge due to its nanometric dimensions, and despite recent remarkable advances[2,3,17–19], its precise morphological and diffusional parameters are still unknown in brain pathology. Here we explored the ECS nanoscale organization in the SN of adult mice after LB-induced neurodegeneration, a model of synucleinopathy for PD. Using state-of-the-art technologies, we revealed a highly heterogeneous ECS displaying localized areas of increased diffusivity and enlarged dimensions in the degenerated SN. We have also demonstrated a link between ECS nanoscale parameters, ECM content and glia. These mice display a degraded hyaluronan matrix in a context of reactive microglia, while experimental matrix disruption modified the pathology and increased ECS width and diffusion. Our results reveal a disorganization of the brain extracellular microenvironment associated to hyaluronan remodeling and microglial activation in the parkinsonian SN.

It has been long established that the ECS volume fraction in the brain ranges between 15 and 20%[1,7], and ECS volume changes have been reported in pathological states[1]. However, estimating its exact width has proven more difficult due to experimental limitations[14]. Here we provide nanoscale ECS dimensions in adult parkinsonian SN analyzed by two different methods: cryofixation-EM and single-nanotube tracking. Both techniques rendered similar distributions of ECS widths (Figs. 1i and 2g), with most values falling around the 100 nm mark and an insignificant proportion of widths over 500 nm. These values are in accordance to a recent report on cryofixed adult mouse cortex[17]. Using STED microscopy, Tønnesen et al. described slightly higher ECS dimensions in organotypic brains slices[18], a preparation that has been shown to display a looser, wider ECS than its adult counterpart[19]. Mapping ECS local width revealed a characteristic channels-and-pools topology, analogous to the interconnected network of sheets-and-tunnels reported by others using 3D-reconstruction models of the ECS[6,41]. We observed a clear trend for larger ECS volume fractions in the parkinsonian SN. Unusually high ECS volume fractions have been reported in the SN of other species[42], thus an alternative method such as real-time iontophoresis with tetramethylammonium (RTI-TMA) should validate these results. Nonetheless, volume fraction is a global parameter that affects -and it is quantified- over an entire region. Local width, on the contrary, provides ECS information at nanometer resolution, a scale more in accordance with biological processes[43]. In this context, we detected increased local dimensions in the parkinsonian ECS. This increase, found both in cryofixed and live SN, is at odds with decreased HA levels, considering that this hygroscopic polymer has been proposed as a determinant of ECS volume[29]. We argue that the changes we observe in ECS dimensions are probably due to the loss of cellular tissue by neurodegeneration, but also the consequence of disrupted cell attachments associated to matrix degradation. Further characterization of cell-ECM interactions conducted at the molecular level are needed to elucidate this phenomenon.

Diffusional parameters are equally augmented in the parkinsonian ECS, although they are not spatially correlated to local

changes in ECS width. Diffusion in the brain ECS has been proposed to be affected by geometry and viscosity[3,44], the latter being mostly determined by macromolecular obstacles. Whether geometric tortuosity or diffusion barriers have differential weight influencing molecular diffusion through the ECS, is still a matter of debate[14]. According to the restricted diffusion theory, SWCNTs in LB-inoculated mice should display reduced geometric hindrance due to increased ECS width. Nonetheless, we present evidence here that regulation of nanoscale diffusivity might not be exclusively imposed by variations of geometric constraints. The comparison between 4MU vs 4MU-LB, with similarly increased diffusion despite larger ECS widths in the latter, suggests that hyaluronan-mediated hindrance plays a major role determining ECS diffusivity. We and others have reported augmented diffusion after ECM removal[2,45]. Here we focus on its most abundant component, HA, a viscous polymer that scaffolds ECM assemblies. Either pathologically-induced (LB) HA degradation as well as chemically-induced (4MU) HA depletion rendered similar increase in nanoscale diffusivity. Thus, we propose HA as a dynamic ECS organizer that determines diffusivity locally. This concept has profound implications: first, spatial regulation of HA, either by removal or overproduction, might establish or modify concentration gradients by locally altering diffusion (i.e., channeling movement of molecules at a nanoscale) either transiently or in a long-lasting manner. This would have repercussions in the specificity of extrasynaptic communication, which is based on local nanoscopic diffusion through the ECS[1,24]. Second, most mathematical models focused on diffusion or bulk flow address the ECS as an empty space. We believe that future ECS models would benefit from including diffusional barriers in addition to topology.

From the vast array of molecules forming the ECM, HA is probably its most versatile component, given its structural and signaling properties[46]. We demonstrate extensive HA degradation in the SN after neurodegeneration. This occurs without affecting CSPGs, as reported in human neurodegeneration[31] or HA loss in vivo[29]. We propose that pathological HA disruption impacts not only diffusion, but also inflammation. Low-MW-HA is a pro-inflammatory molecule[11], and HA breakdown in the SN might trigger the release of additional damage-associated molecular patterns (DAMPs), since matrix metalloproteinases (MMP) activity is increased by fragmented HA[47]. We did not find changes in mRNA levels of neither HA-synthases nor hyaluronidases, but detected increased HA phagocytosis by activated microglia. The significant proportion of HA inclusions found in control mice suggests that microglia might participate in physiological HA turnover. Nonetheless, this balance seems altered in the parkinsonian SN, suggesting the existence of a positive feedback loop microglia-HA. HA internalization is mostly dependent on CD44 receptor[35,48], and its expression is elevated in LB-inoculated mice, similarly to microglial CD44 upregulation in other brain disorders[35], In pathological states, HA can also be degraded non-specifically by reactive oxygen species (ROS)[49], a common feature of neurodegeneration, and we cannot exclude this possibility in our model. The decrease in HA fractal dimension, which we interpret as a loss of interconnectivity, might be due not only to HA degradation but also to proteolysis of HA cross-linkers[32] such as tenascins.

Seed-based models such as the LB-inoculated mouse offer the possibility to study the pathological process at an early stage[50]. While the detailed description of the ECS we provide here is presented as the pathological ECS, our findings have implications to specific mechanisms of PD and other proteinopathies. The obvious hypothesis is that increased ECS diffusion might have a role in the spreading of toxic conformers of α-syn or other proteopathic seeds, and here we show that degrading hyaluronan

in vivo (i.e. Hyase) reduces α-syn inoculum load. This could be due to increased clearance of the inoculum through the ECS, reducing its chance to be internalized by cells, although a quantitative model would better clarify this phenomenon. Degradation of another ECM component (CSPGs) had no effect, ruling out Hyase-mediated indirect ECM disruption as the mechanism. A recent report in *C. elegans* showed that hyaluronidase TMEM2 ameliorates intracellular ER stress modulating pathology[51], suggesting another mechanism for HA breakdown-mediated neuroprotection. In addition, our results suggest that HA modification could play a role via modulation of inflammation. HA fragmentation by Hyase, occurring at the moment of LB-inoculation, is neuroprotective in our model, triggering a substantial microglial activation and reducing proteopathic-seeds load. A plausible explanation is that overstimulation of microglia by HA breakdown exerts a beneficial pro-inflammatory state that impairs seeding. In fact, LMW-HA partially reproduced Hyase-mediated effects in neuroinflammation and α-syn load, and protected from dopaminergic cell loss. Despite we found no evidence that activated microglia engulfs α-syn, we detected Iba1-positive cells surrounding human-α-syn-positive neurons in the murine SN, suggesting that these cells can sense either aggregated α-syn[52,53] or neuronal damage[39]. It is possible that toxic-α-syn-positive cells release injury-related factors such as ATP, which attracts microglia[54]. This would also explain microglia invading the degenerated SNc. Aggregated α-syn can activate microglia[55], upregulate CD44 and enhance migration[56]. In accordance, we observed early inflammation triggered by LB inoculation alone, but to a much lesser extent than in the HA-modified brain, suggesting that HA fragmentation amplifies this phenomenon.

Either at the synaptic cleft, intracellular calcium microdomains, or neurotransmitter receptors, there is an increasing need in neuroscience to refine probing of biological compartments[57]. Studying local changes at the nanoscale, as opposed to global parameters spanning entire regions, is a paradigm shift in neurobiology that finds its latest bastion in the ECS. This is the first study exploring ECS, ECM and glia in an integrative approach within a pathological context, and provides evidence of a dynamic interplay between these players. Rephrasing a 50-years-old hypothesis by ECS research pioneers Schmitt and Sampson[58], we propose the term "brain extracellular microenvironment" to encompass these factors under one unifying concept to facilitate discussions. Finally, the possibility to combine nanoscale ECS exploration with ECM, cellular and subcellular labeling will open new avenues to further understand brain physiology in health and disease.

## Methods

**Animals and surgical procedures**. LB and noLB fractions were purified from fresh frozen post-mortem samples of sporadic PD-patients obtained from brains collected as part of the Brain Donation Program of the Brain Bank GIE NeuroCEB, run by a consortium of Patients Associations: ARSEP (multiple sclerosis), CSC (cerebellar ataxias), France Alzheimer and France Parkinson. The consent documents were signed by the patients themselves or their next of kin in their name, in accordance with the French Bioethical Laws. The Brain Bank GIE NeuroCEB (Bioresource Research Impact Factor number BB-0033-00011) has been declared at the Ministry of Higher Education and Research and has received approval to distribute samples (agreement AC-2013-1887). Human SNpc was dissected and homogenized before a sucrose step gradient[20] and sonicated for 5 min prior to in vivo inoculation to disrupt α-synuclein aggregates into oligomeric and fibrillary fragments. Male 10-weeks-old C57BL/6J mice (Charles River) were unilaterally injected into the *substantia nigra* with 2 μl of either Lewy-body (LB) fractions, containing toxic fibrillary α-synuclein, or noLB fractions, with soluble non-aggregated α-synuclein. Intranigral inoculations were achieved by stereotactic surgery (coordinates from bregma: −2.9 mm anteroposterior, 1.3 lateral, and −4.5 dorsoventral) under deep isoflurane anesthesia at a flow rate of 0.4 μl/min with a 30-gauge Hamilton syringe coupled to a microinjection pump (World Precision Instruments). The needle was left in place for 10 min prior to leakage. Some animals were co-injected with 2 μl of LB fractions and either 1 μl (3U) of hyaluronidase (Hyase) from *Streptomyces hyalurolyticus* (Sigma-Aldrich), 0.025U of chondroitinase ABC (Sigma-Aldrich), low-molecular weight (200 mg/ml) or high-

molecular weight hyaluronan (15 mg/ml, Sigma-Aldrich) or sterile PBS. The Institutional Animal Care and Use Committee of Bordeaux (CE50) approved experiments under license #5520-2016052514328805.

**4-MU treatment**. 6-week-old C57BL/6J mice (Charles River) were fed with normal mouse chow or mixed with 5% (w/w) 4-methylumbelliferone (4MU, Sigma-Aldrich), a concentration reported to effectively inhibit hyaluronan production in vivo systemically[59] and in CNS[60]. The mixture was prepared fresh daily, including water and 33% of cacao to enhance the palatability of the drug. Animals were fed with 4-MU for 4 weeks prior to LB inoculation, and kept on this diet until euthanasia (4 months). Animals were weighted daily during the first month, and weekly afterwards. Despite initial weight loss in the 4MU-treated animals, mice recovered after 2 weeks. Animals showed no sign of stress or particular phenotype, other than weakness of the skin during surgery suture. The Institutional Animal Care and Use Committee of Bordeaux (CE50) approved experiments under license #10721-2017071213284522.

**High-pressure cryofixation and electron microscopy**. Bilaterally LB-inoculated mice were euthanized by cervical dislocation and brains quickly removed and sliced (180 μm thickness) in a VT1200S vibratome (Leica Microsystems) in ice-cold aCSF (see below "acute brain slice preparation" for details). Tissue discs (5 mm diameter) of *substantia nigra* (SN) were immediately transferred to a 200 μm-thick aluminum sample holder (6 mm diameter). Medium was cleared with a filter paper and the cavity filled with 1-hexadecene. The sample was sandwiched with a flat aluminum top plate and high-pressure frozen in a HPM100 device (Leica Microsystems). No more than 10 min elapsed from euthanasia to cryofixation. Samples were kept in liquid nitrogen until further processing. Since the pathology develops in each hemisphere independently, SNs were treated separately as individual samples.

Freeze-substitution resin embedding was carried out in an AFS2 unit (Leica Microsystems). Frozen samples were incubated for 40 h using 0.1% tannic acid in anhydrous acetone at −90 °C, followed by 12 h of 2% OsO₄ in anhydrous acetone at −90 °C. The temperature was then raised to −30 °C with periodic temperature transition gradients of 1 °C/h and to −10 °C by 3 °C/h. Samples were then immersed in Epon resin/anhydrous acetone over 9 h (1/3, 1/1 and 3/1, 3 h each) whilst temperature increased to 20 °C. Finally, samples were added to 100% Epon resin at room temperature for 2 h prior to polymerization at 60 °C for 48 h.

Artifact-free tissue was selected in semi-thin (0.5 mm thick) sections stained with toluidine blue. Ultra-thin sections of selected regions were post-stained with uranyl acetate (1%) and lead citrate and imaged in a transmission electron microscope (H7650, Hitachi) equipped with a SC1000 Orius camera (Gatan).

**Estimation of ECS dimensions from cryofixation-EM images**. ECS was manually segmented from cryofixed-EM images where membranes were easily discernible (Supplementary Fig. 2), using an interactive pen tablet (Wacom) and the TrakEM2 plugin in Fiji/ImageJ (NIH)[61]. Segmented images were binarized, resulting in individual ECS compartments (Fig. 1d). The ECS volume fraction was estimated by dividing the area of ECS compartments by the total area of the image.

To estimate the length of ECS compartments, binary images were skeletonized and the longest shortest path was calculated in ImageJ[62]. ECS local widths were calculated by measuring iteratively the diameter of the largest circle fitting each portion of the ECS compartment, using a 2D implementation of the Local Thickness plugin (ImageJ). The distribution of pixel values obtained from the histograms of ECS local thickness maps was converted to dimensional values using the image pixel size (1.42 nm).

**SWCNTs injection and acute brain slice preparation**. 1 mg of HiPco-synthesized carbon nanotubes (from Rice University) was suspended with 50 mg of monofunctional phospholipid-polyethylene glycol (#mPEG-DSPE-5000, Laysan Bio) in 10 ml of deuterium oxide (Sigma Aldrich). After homogenization for 15 min at 19,000 rpm with a dispersing instrument (IKA T-10 Basic), SWCNTs were further dispersed via tip sonication (20 W for 8 min) and then centrifuged at 800 × g for 60 min to remove nanotube bundles and impurities.

Four months after LB or noLB inoculation, animals underwent a similar stereotactic surgery where 6 μl of SWCNT solution was injected into the cerebral lateral ventricle (coordinates from bregma: −0.5 mm anteroposterior, 1 mm mediolateral, −2.4 dorsoventral). The needle was left in place for 10 min to avoid leakage. Animals received buprenorphine (0.1 mg/kg) as analgesic and returned to their cages. 1 h after SWCNT injection, mice were euthanized by cervical dislocation. Brains were extracted and coronal sections (300 μm thick) were prepared in a Leica VT1200S vibratome. Slicing was performed in ice-cold low-Na⁺ low-Ca²⁺ high-Mg²⁺ artificial cerebrospinal fluid (aCSF) solution containing 180 mM sucrose, 2.5 mM KCl, 1.2 mM NaH₂PO₄, 13 mM glucose, 20 mM HEPES, 10 mM MgSO₄ and 0.5 mM CaCl₂. Slices were then transferred to oxygenated aCSF (gassed with 95% O₂, 5% CO₂, pH = 7.4) containing 124 mM NaCl, 2.5 mM KCl, 1.2 mM NaH₂PO₄, 24 mM NaHCO₃, 13 mM glucose, 3 mM sodium pyruvate, 2 mM MgSO₄ and 2 mM CaCl₂ and kept at room temperature until imaged within 2 h.

**Near-infrared microscopy**. Slices were recorded at 37 °C in a 3D-printed chamber with oxygenated warm aCSF perfused by a peristaltic pump (Fig. 2b). The overall quality of the tissue was initially checked with a standard 4× objective and a white light fiber bundle positioned at 30° above the sample (oblique illumination technique). SWCNTs with a (6,5) chirality were excited at their K-momentum dark exciton vibronic sideband[63] using a 845 nm laser (Coherent) and imaged with a long pass 900 nm filter (Semrock) using a water immersion 60× objective (NA 1.0, Nikon). Images of moving SWCNTs in ECS were collected using a 30 ms exposure time with an EM-CCD camera (Princeton Instruments). Slices were imaged for no more than 1 h. The peristaltic pump was turned off during each recording (2–4 min) to minimize sample drift.

**Single-nanotube localization analysis**. Image analysis was performed using custom MATLAB (MathWorks) and Python scripts. Briefly, individual diffusing SWCNTs in the ECS of live brain tissue were super-localized by fitting images with two-dimensional asymmetric Gaussian functions having arbitrary orientations. Three consecutive images were averaged for each fit to improve the localization precision (~50 nm in water). Slow sample drift was corrected by tracking background features using a variant of the redundant cross-correlation drift estimator[64,65]. Briefly, a region of the image where background features were visible was initially selected for analysis. For each pair of frames $t, u$ ($1 \leq t, u \leq T$ where $T$ is the total number of frames), the cross-correlation was computed on the region of frame $t$ to the region on frame $u$. The maximum of the cross-correlation map estimates the drift between frames $t$ and $u$, $\Delta_{t,u} = r_u - r_t$ (where $r_t$ is the drift at time $t$); the position of this maximum is estimated with subpixel accuracy by quadratic interpolation. Repeating this process over all pair of frames lead to an overdetermined system of $T(T-1)/2$ equations, $\Delta_{2,1} = r_2 - r_1$, $\Delta_{3,1} = r_3 - r_1$, $\Delta_{3,2} = r_3 - r_2$, etc., for the $T$ unknowns $r_1,...,r_T$. As previously described elsewhere[64,65], the outlier equations (i.e. inconsistent equations) were found by comparing the residuals after solving the $T(T-1)/2$-equation system in a least-squares sense (a threshold of 1 px was used for analysis). We found that a more accurate estimation of the drift could be obtained by iteratively repeating the process of solving the overdetermined system and removing outlier equations until the set of equations stabilized. Finally, the resulting drift estimate was smoothed using a local window where the drift estimate on a given frame was weighted according to the number of non-outlier equations.

**Local ECS dimensions and diffusivity from SWCNT images**. Drift-corrected SWCNT coordinates were interconnected to reconstruct individual trajectories. The SWCNT lengths were estimated using the distribution of the longest axis of the 2D asymmetric Gaussian fits when SWCNT movements were undetectable (i.e. displacements between images <40 nm) corrected by the exciton diffusion length (~100 nm)[19]. The length distribution of the analyzed SWCNTs was centered at around 500 nm. Super-resolved images of the ECS were computed on 25 nm pixels by cumulating the SWCNT localizations and convoluting them with a two-dimensional Gaussian of 50 nm full width at half maxima (FWHM) and unit amplitude. Areas of SWCNT exploration were calculated by measuring the binary images of ECS super-resolved maps.

To estimate local ECS dimensions, the shape of the local area explored by individual SWCNTs along their trajectory was analyzed using a 6-points time-window[19]. This time window was chosen at maximum confinement (i.e. when the shape of the local area is maximally distorted by local ECS dimension as compared to expected in unconfined environments) as defined by the eccentricity ratio of the ellipse formed by the SWCNT trajectory (Supplementary Fig. 3). For each trajectory, analysis of the instantaneous mean square displacement ($MSD_{inst}$) was used to estimate the instantaneous diffusion coefficient $D_{inst}$. MSD values were calculated over a sliding window of 450 ms and linear fits were then applied to the first 90 ms to retrieve values of $D_{inst}$. Immobile SWCNTs, characterized by the plateau shape of their global MSD, were excluded from the analysis (Supplementary Fig. 3). Local relative diffusivity ($D_{inst}/D_{ref}$)[3,19] was obtained by comparing $D_{inst}$ with the free diffusion of the considered SWCNT in a fluid having the viscosity of the cerebrospinal fluid ($\eta_{ref}$):

$$D_{ref} = \frac{[3k_B T \ln(2\varphi)]}{8\pi\eta_{ref}L} \quad (1)$$

where $k_B$ is the Boltzmann constant, $T$ is the temperature of recording, $\varphi$ is the SWCNT aspect ratio, and $L$ is the nanotube length. Spatial diffusivity maps were created by associating local relative diffusivity values with super-localized centroid positions. Values falling within the same pixel (25 nm) were averaged and convoluted with a 2D Gaussian of 50 nm FWHM and unit amplitude.

**Immunostaining and confocal imaging**. Mice were anesthetized with pentobarbital (100 mg/kg ip) and intracardially perfused with 0.9% saline followed by ice-cold freshly-prepared 4% paraformaldehyde (PFA) in 0.1 M PB. Brains were post-fixed for 3 h in 4% PFA and 40 μm-thick vibratome coronal sections were collected. For visualization of the hyaluronan matrix or CSPGs, slices were blocked with 1% bovine serum albumin (BSA) in PBS with 0.1% saponin for 1 h, followed by streptavidin-biotin blocking kit (Vector Labs) for 20 min, and incubated overnight with either biotynilated-hyaluronic acid binding protein (HABP from bovine

nasal cartilage, Merck-Millipore) or biotynilated Wisteria Floribunda Agglutinin (WFA, Vector Labs) diluted in blocking solution. Staining was revealed with Streptavidin-Atto647N (Sigma-Aldrich), with minimum non-specific labeling due to appropriate blocking of endogenous biotins (Supplementary Fig. 5c). Double or triple labeling was achieved by re-blocking samples with 4% normal goat serum and overnight incubation with primary antibodies for the following antigens: Iba1 (019-19741, Wako), CD68 (FA-11, Bio-Rad), TH (LNC1, Merck-Millipore), human α-synuclein (Syn211, Thermo Fisher). Staining was revealed with appropriate secondary antibodies conjugated with Alexa 488 or 594 (Thermo Fisher) and incubated 30 min with nuclear dye Hoescht 33342 (10 μM, Thermo Fisher). Sections were mounted on #1.5 coverslips with Mowiol + DABCO and left to dry overnight in darkness. Confocal images were obtained in a Leica TCS SP8 microscope, maintaining image acquisition settings (laser power, AOTF, detection parameters) between sessions. Image stacks (pixel size ~100 nm, z-step 0.5 μm) were acquired with 20X or 63X Plan Apo CS objectives with oil immersion.

**Analysis of hyaluronan network and microglia**. HABP, Iba1, CD68 and human α-syn levels were quantified on 12-bit ×63 confocal images acquired from coronal SN sections, at least 100 μm rostral or caudal to the point of injection. At least 4 slices were quantified per animal (2 rostral and 2 caudal). The area of staining was calculated by automatic thresholding in Fiji/ImageJ and related to the total area of the image. Iba1-positive somas were manually counted in ROIs based on the TH channel and normalized by surface area. For CD68, a mask generated in the Iba1 channel was applied, to ensure quantification of CD68 labeling exclusively inside microglia. For α-syn levels, the mean gray value of the contralateral side (where no injection was performed, therefore, no human α-syn staining was possible) was deemed as background and set as value = 0 for the quantification in the ipsilateral side.

To quantify hyaluronan network complexity by analysis of fractal dimension, HABP-TH double immunofluorescence images were segmented in the HABP channel by thresholding, then binarized and skeletonized in Fiji. Analysis was spatially restricted by creating a mask in the TH channel and therefore quantifying only the interstitial matrix of the SNc, where interconnectivity is more evident. Box counting fractal dimension was calculated with the FractalCount plugin for Fiji/ImageJ using default parameters. The length of hyaluronan cables was estimated on skeletonized images by quantifying the longest shortest path of the skeleton.

Hyaluronan engulfment by microglia was quantified manually in images where Iba1-positive somas were visible in the entirety of the z-stack, and expressed as % of Iba1-positive cells with hyaluronan inclusions. To ensure quantification of only sizable HABP-positive clusters inside Iba1 cells, background in the HABP channel was subtracted and a median filter of 1px was applied to eliminate small staining artifacts. Colocalization, fluorescence profiles, orthogonal projections, and maximal intensity projections were performed in Fiji/ImageJ (NIH). Images were linearly adjusted for brightness and contrast for visualization purposes, applying similar values to control-treated or contralateral-ipsilateral samples.

**Unbiased stereological cell counts**. For SN cell counts, every fourth section was processed for TH immunohistochemistry and counterstained with cresyl violet, resulting in 6 sections to be counted per animal. Briefly, 40 μm free-floating coronal sections were blocked 1 h with 1% BSA + 0.3% Triton X-100 in PBS and incubated overnight with anti-TH (clone LCN1, Merck-Millipore). Staining was revealed using anti-mouse HRP EnVision kit (Dako) for 30 min followed by DAB staining (30 s).

Stereology was performed using the Mercator Pro software (Explora Nova) coupled to a Leica DM-6000B microscope with a motorized XYZ stage. The number of neurons in the SN was estimated using the optical fractionator method and calculated based on the following formula: $N = \Sigma Q- \times 1/ssf \times 1/asf \times t/h$, where $N$ is the estimate of the total number of cells, $\Sigma Q-$ is the number of objects counted, $ssf$ is the section sampling fraction, $asf$ is the area sampling fraction, and $t/h$ is the actual section thickness divided by the height of the dissector.

**qPCR**. Nigral samples were homogenized in Tri-reagent (Euromedex) and RNA was isolated using a standard chloroform/isopropanol protocol[66]. cDNA was synthesized from 2 μg of total RNA using RevertAid Premium Reverse Transcriptase (Fermentas) and primed with oligo-dT primers (Fermentas) and random primers (Fermentas). qPCR was performed using a LightCycler 480 Real-Time PCR System (Roche). qPCR reactions were done in duplicate for each sample, using transcript-specific primers, cDNA (4 ng) and LightCycler 480 SYBR Green I Master (Roche) in a final volume of 10 μl. The PCR data were exported and analyzed using an informatics tool (Gene Expression Analysis Software Environment) developed at the NeuroCentre Magendie. For the determination of the reference gene, the Genorm method was used[67]. Relative expression analysis was corrected for PCR efficiency and normalized against two reference genes: gapdh and ubc. The relative level of expression was calculated using the comparative (2 −ΔΔCT) method and expressed as fold-change. Primer sequences are detailed in Supplementary Table 1.

**Thioflavin T assay**. To assess the level of aggregation of LB upon contact with Hyase, we performed a Thioflavin T assay[68]. Briefly, 5 μl of sonicated LB with or

without 2 U of Hyase were added to 495 µl of a solution containing 20 mM ThT, 50 mM glycine in water, pH 8.5. Freshly-prepared amyloid-beta was left overnight at room temperature to oligomerize and used as positive control for aggregation. Fluorescence was measured every hour in a FLUOstar Optima (BMG Labtech) at a 450/480 nm excitation-emission wavelengths.

**Proteinase-K assay.** To analyze α-synuclein aggregates in the SN of LB(Hyase)-inoculated mice we performed a Proteinase-K assay, an enzyme that eliminates only non-aggregated proteins. Midbrain slices were incubated with 0.1 µg/ml of Proteinase-K (Sigma-Aldrich) in agitation during 10 min at room temperature and washed. We next performed standard immunohistochemistry using a pan (rodent + human) α-synuclein antibody (syn-1, clone 42, BD Biosciences) and revealed with HRP EnVision Kit (Dako) and DAB. No-PK slices were processed in parallel as a positive control for α-synuclein immunohistochemistry.

**ELISA.** Hyaluronan content was quantified in blood samples from Ctrl and 4MU-treated mice using the Hyaluronan Quantikine ELISA Kit (R&D Systems) according to manufacturer instructions. Briefly, blood was extracted from mice by cardiac puncture with a 1 ml syringe primed with EDTA 15% and transferred to polypropylene tubes. Plasma was separated by centrifugation and stored at −80 °C until further use. Hyaluronan content was determined in 40x dilutions of plasma sample by colorimetric ELISA in a FLUOstar Optima (BMG Labtech) plate reader. Results were interpolated into a standard curve using a 4-PL curve fit corrected for background absorbance.

**Statistics and reproducibility.** Methodologies that are routinely used in our laboratories were successfully reproduced according to our previous publications: LB-induced neurodegeneration, single-nanotube tracking, hyaluronan, and microglia analysis. For newly acquired methodologies (cryofixation-EM, 4MU treatment, Hyase and ChABC injection), we conducted pilot experiments to validate those methods, which were successfully replicated in this study. The key finding of increased ECS width and diffusion in LB-inoculated mice was replicated in two completely different experiments performed independently that achieved the same conclusion. Statistical analyses were performed in GraphPad Prism 7.0 software or MATLAB. Datasets were initially tested for normal distribution with D'Agostino-Pearson normality test. Normal populations were analyzed by unpaired two-tailed *t*-test (paired for contralateral vs ipsilateral comparisons) or One or Two-way ANOVA followed by Holm–Sidak post-hoc test, depending on the number of variables. Continuous datasets (i.e. with a large number of data points) with non-normal distributions were analyzed by Kolmogorov–Smirnov (KS) test comparing cumulative distributions. For all statistical tests, the level of significance was set to $P < 0.05$. Exact $P$ values are reported in each figure. In KS tests for large datasets, an approximate $P$ value was calculated. The number of replicates ($n$) is reported in figure legends. Discrete data are represented as mean ± SEM unless specified otherwise. Continuous datasets are represented as relative frequency or cumulative distributions.

**Availability of unique biological samples.** LB and noLB fractions from PD patients are obtained from limited material. The authors would gladly extract them from the provided samples.

**Reporting summary.** Further information on experimental design is available in the Nature Research Reporting Summary linked to this paper.

## Data availability
The data that support the findings of this study are available from the corresponding authors upon reasonable request. Source data are provided with this paper.

## Code availability
Customs scripts and software code generated for this paper are available from the corresponding authors upon reasonable request.

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

## Acknowledgements

We thank Guillaume Dabée, Melissa Deshors, and Elisabeth Normand for animal care, Melisa Petrel and Sabrina Lacomme for technical assistance in cryofixation, and Thierry Leste-Lasserre from the genomic platform at Neurocentre Magendie for qPCR analysis. We also thank Olatz Pampliega and Amanda Sierra for helpful discussions. This work was supported by grants from Agence Nationale de la Recherche (ANR-14-OHRI-0001-01, ANR-15-CE16-0004-03), IdEx Bordeaux (ANR-10-IDEX-03-02) and Labex Brain (ANR-10-LABX-43). F.N.S. acknowledges postdoctoral grants from Basque Government (POS_2016_1_0098) and Spanish Ministry of Science and Innovation (IJCI-2017-32114). C.P. acknowledges funding from EU's Horizon 2020 research and innovation program under the Marie Skłodowska-Curie grant No 793296. University of Bordeaux and CNRS provided infrastructural support. Cryofixation, electron microscopy and confocal imaging were performed at the Bordeaux Imaging Center, a service unit of CNRS-INSERM and University of Bordeaux, member of the national infrastructure France BioImaging, granted by ANR-10INBS-04-0. Human samples were obtained from Brain Bank GIE NeuroCEB (BRIF number 0033-00011), funded by the patients' associations France Alzheimer, France Parkinson, ARSEP, and "Connaître les Syndromes Cérébelleux", to which we express our gratitude.

## Author contributions

F.N.S., C.P., A.K.M., and N.D. carried out single-nanotube tracking experiments. C.P. and A.L. performed SWCNT analysis. F.N.S. and E.D. performed and analyzed cryofixation-EM experiments. F.N.S., M.L.A., and P.G. analyzed histopathology. B.D. purified LB fractions. F.N.S. and M.L.A. performed surgeries. F.N.S., B.D., L.G., L.C., and E.B. conceptualized the study and interpreted the results. F.N.S. wrote the paper with input from all authors. L.G., L.C., and E.B. secured funding and coordinated the project.

## Competing interests

E.B. is Chief Scientific Officer of Motac Neuroscience Ltd. All other authors declare no competing interests.
