## [Peer Review File · Nature Communications]

Reviewers' comments:

Reviewer #1 (Remarks to the Author):

This paper uses cryo-fixed electron microscopy in combination with super-resolution tracking of single-walled carbon nanotubes (SWCNTs) to study the extracellular space (ECS) in normal and alpha-synuclein induced neurodegeneration in the mouse substantia nigra (SN). Using several manipulations, including Lewy body (LB) injection, hyaluronidase and chronic administration of 4-methylumbelliferone, together with a variety of histological and analytical approaches, it is shown that the hyaluronan (HA) component of the extracellular matrix (ECM) plays a major role in the re-organization of the ECS under these pathological conditions.

The paper represents an important further advance in the exploration of the ECS on the nanoscale and may contribute to clarifying some of the mechanisms that underlie Parkinson's disease. The use of SWCNTs is an emerging technique and is revealing information about the ECS that cannot be obtained with the older diffusion methods. There may be opportunities to extract additional information in the present study by comparison with earlier studies measuring diffusion on a macroscopic scale.

The techniques are in general use but the super-resolution tracking of SWCNTs would only be possible in a few laboratories at this time.

Major points

This paper employs a multifaceted attack on the nanoscale organization of the ECS. The most novel aspect is the use of SWCNTs to directly probe the diffusion characteristics of the ECS. This approach has been used by some of the present investigators in two other recent publications. Cryo-fixation for electron microscopy of the ECS has been employed by other groups in a few studies but only now is becoming a fairly routine procedure. The numerous other techniques are established and appropriate. Thus one may see the study as having three aspects (a) continuing validation of the SWCNT technology (b) a study of the effect of HA modification on ECS structure (c) a contribution to understanding the increasingly complex etiology of Parkinson's and Lewy Body diseases. This review will focus mainly on topic (a).

The SWCNT method is yielding important new insights into the ECS and this study is a further valuable contribution. The authors show that both cryo-fixed EM and SWCNT give comparable distributions for the width of the ECS and cryo-fixed EM also indicated that the volume fraction in substantia nigra is about 20%. Most measurements of ECS volume fraction in the brain with macroscopic methods, such as the Real-Time Iontophoresis (RTI) method, do give values around 20% so the present results would be validated. But the volume fractions of the SNc and SNr have been measured in guinea pig brain slices using the RTI method (Cragg et al., *J. Neurophysiol.* 85: 1761-1771, 2001) and in both regions volume fraction was unusually high at 30%. Perhaps this value is unique to the guinea pig; however the same study found a volume fraction of 22% in the guinea pig cortex, which is comparable to values in rat and mouse. Questions will remain until RTI measurements in the SN of the mouse are available but in the meantime: How confident are the present investigators in their estimates of volume fraction?

A second feature revealed by the SWCNT measurements is that the median relative diffusivity in the normal SN is 0.026 and that in the LB-exposed SN is 0.042 (page 7). Translating these values into tortuosity for comparison with prior studies one obtains tortuosity values of 6.2 and 4.9 whereas the tortuosity obtained in the normal guinea pig SNc and SNr were 1.58 and 1.69 respectively (see Cragg reference above). For comparison the guinea pig study found a tortuosity of 1.59 in the cortex. Crucially, those tortuosity values were obtained with the small tetramethylammonium molecule and are similar to numerous other measurements in brain tissue with this molecule (see Ref. 1). The huge values of tortuosity seen in the present study are only

comparable to the value obtained using the Integrative Optical Imaging (IOI) method with a quantum dot of 35 nm diameter in the rat cortex (Thorne & Nicholson, PNAS 14: 5567-5572, 2006). The quantum dot tortuosity was interpreted using restricted diffusion theory to indicate significant interaction of the particle with the walls of the ECS and this view was consistent with the observation that tortuosity rises as a function of molecular diameter for a range of macromolecules (see Ref. 7). Assuming the free diffusion values have been calculated correctly here (page 21), there is no doubt that the large tortuosities (small relative diffusivities) obtained with SWCNTs represent major hindrance of the nanotube compared to a typical small molecule. If this hindrance arose from interaction with walls of the ECS one might expect the values to decrease with an expanded ECS, which is reported here. Alternatively, the hindrance might be caused by interaction with the ECM and then the hindrance following modifications to the HA could clarify the situation. There may be an opportunity here for some useful further consideration of the present data.

There is also limited comment on resolution. Figs. 2F and 6e have colored calibration bars that go down to 0 nm yet there has to be a practical limit to the width of a channel that SWCNT can explore. This is more pertinent given the large hindrance experienced by the nanotubes which suggests there could be regions of the ECS that nanotubes are unable to access.

The manipulations of HA collectively make a convincing case for an important role for this glycosaminoglycan in structuring the ECS. Exactly how it does so is still to be determined. As the authors note, the present results do not completely accord with the recent study of Arranz et al (Ref 39), which reinforces the point that the elucidation of mechanisms is just beginning and these nanoscale tools are likely to play a major role. It is very important therefore to continue to refine and validate these tools.

The relation of the ECS to Parkinson's disease is more tenuous. The enhanced spread of toxic proteins may indeed be a factor (page 14) but it is difficult to estimate its importance without a quantitative model. Over the years, many studies have shown changes in ECS properties that correlate with pathophysiological manipulations (see Ref. 1) but strict causality has remained elusive.

Minor points

Title: Possibly change to: "Synucleinopathy in substantia nigra alters..."

Abstract. Line 26. "reveal" may be better than "divulge".

Page 3, line 49. Omit "prevailing".

Page 3, line 51. Change "yet" to "still".

Page 3 lines 46 - 47 and 52. A major function of the ECS is to provide a reservoir of ions to support the membrane potential and action and synaptic potentials. Thus the major constituent is probably NaCl.

Page 3 line 56. It is not clear that the ECM functions as a "space-filler" because large macromolecules and quantum dots can diffuse through it. The precise form taken by the matrix is unknown but it might more resemble a "hydro-gel". It may have a role in maintaining the patency of the ECS.

Page 4, line 79 "a first".

Page 4, line 84. Substitute "integrated" for "integral"?

Page 5 line 100 may be better to say "...difficult to implement...".

Page 5 line 116. See comment above about volume fraction of ECS in SN.

Page 7, Line 168-170. It is not clear what aspect of "diffusion" is being compared here. Elsewhere it appears that there is a correlation between ECS dimensions and diffusion. The use of the word "anisotropy" here may be confusing; see remarks about anisotropy elsewhere in this review.

Page 10 line 242. Define "PK".

Page 10, line 252. Replace "respect" with "in comparison to"?

Page 11 line 274. Change "alter" to "alters"

Page 12 line 296. "live brain slice"

Page 12 line 301. Not clear what the authors want to signify by "founding stone". "...a foundation for...?"

Page 12 line 304. Volume fraction in pathological states is more extensively described in Ref 1.

Page 12 line 313. Probably "analogous" better than "analog".

Page 12 line 313. The phrase used in Refs. 6 and 37 was "sheets and tunnels".

Page 13, line 327. If we define anisotropy as being a macroscopic property characterized by a six-component symmetric tensor representing the effective diffusion coefficient (usually assume to have only three components because measurements are taken in the principal axes) then the brain is not always anisotropic. Much of the cerebral cortex is isotropic (hence its popularity for measurements); verified examples of anisotropy occur in the molecular layer of the cerebellum and the corpus callosum. It might be possible but challenging to measure anisotropy on the nanoscopic scale. What is evident here perhaps is local inhomogeneity or channeling on a scale of a few micrometers.

Page 13 line 328. The more recent contribution about viscosity of the ECS from the Rusakov lab might also be cited here (Ref. 3).

Page 13 lines 331-332. This is a subtle issue. If the hindrance is largely caused by interaction with the ECS boundaries, as may be the case for SWCNTs, then restricted diffusion theory says that relative diffusivity will increase as the walls move further apart. On the other hand, a small molecule will have negligible interaction the walls and an increase in ECS width may only bring about a small change in relative diffusivity. Consider the very simple case of an ECS formed by an ensemble of cubic cells (Tao & Nicholson, J. Theoret. Biol. 229: 59 – 68, 2004). The relative diffusivity for a vanishingly small molecule is given by $2/(3 - \alpha)$. Assuming that alpha tracks ECS width then if alpha goes from 0.2 to 0.3, representing a 50% increase, relative diffusivity only goes from 0.71 to 0.74 – a 4% increase.

Page 13 line 343. For the reason noted above, there may be some confusion about the use of the word "anisotropic" here. The anisotropic diffusion described in both of the cited references (Refs. 1 and 23) was of the macroscopic variety. Here it appears that the authors are talking about a local channeling of substances in the ECS. Note that extrasynaptic communication is not dependent on (macroscopic) anisotropic diffusion. It could equally well occur in a macroscopically isotropic medium. Even in a strongly anisotropic tissue, such as the molecular layer of the cerebellum, molecules do diffuse in all three axes. Bulk flow in the ECS (if it occurs) probably will be vectorial but even then there will be a diffusive component perpendicular to the flow. It is, however, intriguing to speculate about how HA or other ECM components might channel molecular movement on the nanoscopic scale.

Page 14, line 353 Change "found" to "find".

Page 15 line 387. There is perhaps no need to introduce the term "extracellular compartment" but rather to go back 50 years to a remarkable volume in the Neuroscience Research Program Bulletin series entitled "The Brain Cell Microenvironment" (Schmitt and Sampson, NRP Bull. 7: 277 – 417, 1969). Sadly, this publication is no longer easily available. This volume was way ahead of its time in focusing on the role of the ECM in brain function and coining the term "brain cell microenvironment". This term has been used in a number of papers about the ECS. Perhaps "brain extracellular microenvironment" might be even better.

Page 18 line 471 Change "and interactive" to "an interactive".

Page 19. What was the length distribution of SWCNTs?

Page 21 lines 533-534. "...equations in which that frame participates."?

Figure 1. Panels f and g: ordinates mislabeled – they are cumulative frequency.

Figure 1. Panel e: the volume fraction is based on $n = 3$ mice which seems a bit low for a statistical test. The actual measuring procedure in each mouse presumably takes place over many sections so the n value may be a bit misleading. Identifying the ECS in electron micrographs is sometimes challenging.

Figure 1. Here in the histograms in panel i and also in Figure 2 panels d and g, as well as Figure 6, panels g and i, it is difficult to see the rose-colored bars against the gray.

Figure 2. Panels c and d. Presumably the short distance over which trajectories are imaged – a few

micrometers – ensures that the nanotubes remain in or close to the focal plane of the microscope, so no nanotubes are lost by going out of the plane. In panel f the dimension scale goes to 0 nm – what is the lower limit of the method? There are some indications in the text that it is around 50 nm. The statement about anisotropy should be considered in the light of earlier comments in this review.

Figure 3. Ordinate in panel d mislabeled: it is cumulative frequency.

Figure 5. Second line perhaps should read "... lasts only a few days...".

Figure 6. Again panel e shows 0 nm scale bar value.

Reviewed by:

Charles Nicholson

Professor Emeritus
NYU School of Medicine

Reviewer #2 (Remarks to the Author):

This is a potentially important, ground-breaking body of work. The authors address a hitherto underappreciated aspect of Parkinson's disease; the role of extracellular space and matrix (ECS and ECM) in the aetiology of the synuclein-driven neurodegenerative process in the substantia nigra. It is also a strength that they report a successful attempt to reverse the neurodegenerative process by interfering with these ECM mechanisms, so suggesting the potential therapeutic importance of the findings.

The work employs a multi-disciplinary set of techniques to address this issue from multiple directions. The techniques employed, from animal model to analysis of structure and diffusion properties etc of ECM are the most appropriate available, being state-of-the-art and avoiding many limitations of alternative approaches.

Many of the findings appear robust and overall the authors present a compelling argument to support their hypothesis. Thus, the data on, for instance, width and diffusion maps of the extracellular space, microglial activation, the degradation of the hyaluronan matrix, cytokine levels and neuroprotective effect of hyaluronidase are compelling. In themselves these are sufficient to make the manuscript of interest and could be highlighted better in the Abstract.

However, a weakness of the work is that not all measures are evaluated with the same level of rigour. Some aspects of the work seem underpowered, and so conclusions are uncertain, at other times, the extrapolation of conclusions to broader populations of animals is uncertain. Specifically,

- Fig 1e, a trend towards an increase in %ECS in Lewy body treated animals is seen. The analysis appears to only employ N=3, if a larger group were used, it is likely that a clearer conclusion could have been drawn. If this were significant, it would be a simple and easily appreciated measure to open the argument for the role of ECS changes in PD.

- In the K-S analyses, e.g. Fig 1f and g, and elsewhere; although several hundred datapoints are entered into the analyses, these only come from very small numbers of animals in each treatment group, for instance N=3 in the case of Fig 1f. While the significance suggests that it can be stated that the 3 animals in one group likely had a distribution dissimilar to the 3 animals in the other group, this does not mean that it is so likely that populations of animals treated in this way would be significantly different. The sample is a sample of the two groups of three animals not a sample of the two treatment populations, and thus the statistics are powered for, and assess that comparison.

- The data on the ECM structure in the face of chronic HA reduction are very interesting and could add significantly to the ability to draw conclusions as to cause and effect of ECS dysfunction in neurodegeneration. However, the authors relegate the data showing the effect of this chronic HA reduction on neurodegeneration to the Supplementary materials. This is understandable as the data are inconclusive, but again suffer from low N, N=4. If this aspect of the work were properly

powered to show a change in neurodegeneration, and a clear effect of treatment were seen, or not, then the conclusions to be drawn would be stronger.

The authors use the human-derived Lewy body extracts to induce synucleinopathy. This is a reasonable choice and one of several available models that can produce robust synucleinopathy and degeneration in animals. However, one limitation of the model lies in the ability of the community to reproduce the findings reported. This limitation stems mainly from the availability of LB extract, either from the same source as used in these studies or from other sources that can be prepared in a way that that produces verifiably-similar protein material. It would thus have been a significant enhancement of the work had they shown replication of some of the key findings with another model of synucleinopathy. Of course, an orthogonal approach to modelling PD, would have further demonstrated the robustness, and likely translatability, to the disease, of the conclusions.

Reviewer #3 (Remarks to the Author):

Soria et al have used electron microscopy in cryofixed tissue and single nanotube tracking in live brain tissue to study extracellular space (ECS) in healthy and pathological adult mouse brain. They show that alpha synuclein-induced neurodegeneration increased the length and width of local "pools" of ECS, increased the level of nanoscale diffusion and was linked to the levels of hyaluronan. The results and conclusions are clearly presented, and the findings are novel and of interest to many in the field.

The manuscript would benefit from the suggested clarifications below, especially regarding the role of hyaluronan in ECS size.

Major comments:

1. Another ECM component should be analysed alongside hyaluronan to ensure that the effect observed is specific to hyaluronan and not a general effect on the ECM as a whole. This should include staining and quantification of other ECM components after synuclein-induced neurodegeneration and the use of another ECM-degrading enzyme for the Hyase experiments. Currently, the control used is injection of PBS. Injection of an ECM-degrading enzyme that targets another component of the matrix present would show if the neuroprotective effects observed are due to hyaluronan levels, or simply ECM remodelling.
2. The authors state that small MW-HA or hyaluronan breakdown has a neuroprotective effect, as injection of HMW-HA did not show an effect. LMW-HA/HA fragments should also be injected to see if the presence of LMW-HA is sufficient for the neuroprotective effect, or if the endogenous HMW-HA must be degraded.
3. No neuroprotective effect was shown for 4-MU at 4 months, unlike Hyase. Is there a neuroprotective effect of 4-MU at an earlier time point? That is later lost due to a decrease in Iba1+ microglia levels over time? Levels of microglia activation should be analysed at an earlier stage (when 4-MU would be expected to start reducing HA levels) to assess if 4-MU alters microglia levels/activation as Hyase did at 72 h. The levels of other ECM components present in the matrix should also be analysed in these animals to see if that may explain the lack of neuroprotection, as should the levels of the Hyaluronidases.

Minor comments:

1. The authors state that the width and length of local ECS "pools" is increased, but the total volume of ECS space is not. This is a little contradictory. The total number of pools should also be quantified, as the images in Fig. 1h appear to show there are fewer, larger "pools" present after synuclein-induced neurodegeneration.
2. There are several methodological details that should be included in the results. How long was

the imaging of the SWCNTs performed for? How was the mRNA expression analysed? Including this information in the results as well as the methods will enable the reader to follow the experiments outlined more easily.

3. Microglia levels appear to have been quantified by the area of Iba1 staining (Fig 4c), yet Iba1+ soma are used for another quantification (Fig 4f). Iba1+ soma should also be analysed when the level/number of microglia are quantified (Fig 4c, 5j).

4. Supplementary videos of SWCNTs in synuclein-induced neurodegeneration brain tissue and the Hyase condition should also be included.

Synucleinopathy alters nanoscale organization and diffusion of the brain extracellular space through hyaluronan remodeling

Response to reviewers

REVIEWER 1

1. The SWCNT method is yielding important new insights into the ECS and this study is a further valuable contribution. The authors show that both cryo-fixed EM and SWCNT give comparable distributions for the width of the ECS and cryo-fixed EM also indicated that the volume fraction in substantia nigra is about 20%. Most measurements of ECS volume fraction in the brain with macroscopic methods, such as the Real-Time Iontophoresis (RTI) method, do give values around 20% so the present results would be validated. But the volume fractions of the SNc and SNr have been measured in guinea pig brain slices using the RTI method (Cragg et al., J. Neurophysiol. 85: 1761-1771, 2001) and in both regions volume fraction was unusually high at 30%. Perhaps this value is unique to the guinea pig; however the same study found a volume fraction of 22% in the guinea pig cortex, which is comparable to values in rat and mouse. Questions will remain until RTI measurements in the SN of the mouse are available but in the meantime: How confident are the present investigators in their estimates of volume fraction?

We thank the reviewer for this thoughtful comment. We believe that the measurement of the volume fraction by diffusion (RTI-TMA) is inherently different than the method we used in Fig. 1 based solely on geometry by cryofixation-EM. Cryofixation-EM, in the way we have implemented it, is a 2D approximation that does not take into account diffusional barriers (the strength of the method is that provides nanometric resolution), hence the results might differ from RTI-TMA. As noted by the reviewer, the distribution of ECS widths calculated from cryofixation-EM data are very similar to the ones estimated from the nanotube analysis in live tissue, and we believe that this highlights its accuracy. We have modified the discussion stating the limitations of the cryofixation-EM method to estimate the ECS volume fraction in the SN and highlighted the need to validate these results by a diffusional method such as RTI-TMA. We have included the reference by Cragg 2001 to inform the reader that different α values for the SN exist in other species.

2. A second feature revealed by the SWCNT measurements is that the median relative diffusivity in the normal SN is 0.026 and that in the LB-exposed SN is 0.042 (page 7). Translating these values into tortuosity for comparison with prior studies one obtains tortuosity values of 6.2 and 4.9 whereas the tortuosity obtained in the normal guinea pig SNc and SNr were 1.58 and 1.69 respectively (see Cragg reference above). For comparison the guinea pig study found a tortuosity of 1.59 in the cortex. Crucially, those tortuosity values were obtained with the small tetramethylammonium molecule and are similar to numerous other measurements in brain tissue with this molecule (see Ref. 1). The huge values of tortuosity seen in the present study are only comparable to the value obtained using the Integrative Optical Imaging (IOI) method with a quantum dot of 35 nm diameter in the rat cortex (Thorne & Nicholson, PNAS 14: 5567-5572, 2006). The quantum dot tortuosity was interpreted using restricted diffusion theory to indicate significant interaction of the particle with the walls of the ECS and this view was consistent with the observation that tortuosity rises as a function of molecular diameter for a range of macromolecules (see Ref. 7). Assuming the free diffusion values have been calculated correctly here (page 21), there is no doubt that the large tortuosities (small relative diffusivities) obtained with SWCNTs represent major hindrance of the nanotube compared to a typical small molecule. If this hindrance arose from interaction with walls of the ECS one might expect the values to decrease with an expanded ECS, which is reported here. Alternatively, the hindrance might be caused by interaction with the ECM and then the hindrance following modifications to the HA

could clarify the situation. There may be an opportunity here for some useful further consideration of the present data.

We thank the reviewer for this thorough comment. Indeed, what we observed in Fig. 6d is a significant change in the diffusivity values after the HA depletion, which is only poorly reflected in the dimensional analysis (Fig. 6f). This data demonstrates that HA depletion was mainly responsible for the change in diffusivity values detected with the SWCNTs. We added a comment on this in the results section, page 11: “ECS widths were also increased in both 4MU and 4MU+LB. However, the increase in comparison to control SN was greater when the LBs were present (median=135 nm Ctrl, 154 nm 4MU, 195 nm 4MU-LB; Fig. 6e, h, i), suggesting that neurodegeneration has more impact on ECS morphology (widths), while ECS diffusivity depends mostly on HA fluctuation (compare Figs. 6f and h)”. The same considerations were already highlighted in the discussion of the manuscript in several comments, but we have made this more explicit on page 13: “As demonstrated in the comparison between 4MU vs 4MU-LB, with similarly increased diffusion despite larger ECS widths in the latter, hindrance by the ECM plays a major role determining ECS diffusivity”.

We refrain from including any comment on tortuosity values. Indeed, as the reviewer pointed out as well, tortuosity values depend on the hydrodynamic dimension of the diffusing object and are therefore difficult to compare.

3. There is also limited comment on resolution. Figs. 2F and 6e have colored calibration bars that go down to 0 nm yet there has to be a practical limit to the width of a channel that SWCNT can explore. This is more pertinent given the large hindrance experienced by the nanotubes which suggests there could be regions of the ECS that nanotubes are unable to access.

Thanks for pointing out this error. Scale bars in the SWCNT figures have been changed to reflect the 50 nm resolution limit reached with the SWCNT method.

4. The manipulations of HA collectively make a convincing case for an important role for this glycosaminoglycan in structuring the ECS. Exactly how it does so is still to be determined. As the authors note, the present results do not completely accord with the recent study of Arranz et al (Ref 39), which reinforces the point that the elucidation of mechanisms is just beginning and these nanoscale tools are likely to play a major role. It is very important therefore to continue to refine and validate these tools.

We appreciate this comment. We agree with the reviewer that this point reveals that the picture is not easily discernible and more refinement of ECS exploration is needed. We also believe that difference between both studies might arise from the different brain region, mice age and resolution used. We concur also that SWCNTs (and other superresolution tools) can play a major role in the characterization of the ECS environment and need further refinement and validation.

5. The relation of the ECS to Parkinson’s disease is more tenuous. The enhanced spread of toxic proteins may indeed be a factor (page 14) but it is difficult to estimate its importance without a quantitative model. Over the years, many studies have shown changes in ECS properties that correlate with pathophysiological manipulations (see Ref. 1) but strict causality has remained elusive.

We thank the reviewer for this clarification. We have added “...even though a quantitative model would better clarify this phenomenon.” to our discussion about the role of ECS in the spreading of proteopathic seeds. We believe that the role of the ECS in these pathologies has more to do

with inflammation (the ECM as a damage-associated signal) and perhaps a role in pathophysiology through alteration of synaptic transmission or ER stress, all ideas stated several times in the text.

Minor points

We thank the reviewer for this thorough list and careful revision of the phrasing. We have made all the necessary changes in the text (highlighted in yellow) and figures, including:

6. If we define anisotropy as being a macroscopic property characterized by a six-component symmetric tensor representing the effective diffusion coefficient (usually assume to have only three components because measurements are taken in the principal axes) then the brain is not always anisotropic. Much of the cerebral cortex is isotropic (hence its popularity for measurements); verified examples of anisotropy occur in the molecular layer of the cerebellum and the corpus callosum. It might be possible but challenging to measure anisotropy on the nanoscopic scale. What is evident here perhaps is local inhomogeneity or channeling on a scale of a few micrometers

“Anisotropy” has been replaced all along the text by “inhomogeneity” or “local channelling”.

7. This is a subtle issue. If the hindrance is largely caused by interaction with the ECS boundaries, as may be the case for SWCNTs, then restricted diffusion theory says that relative diffusivity will increase as the walls move further apart. On the other hand, a small molecule will have negligible interaction the walls and an increase in ECS width may only bring about a small change in relative diffusivity. Consider the very simple case of an ECS formed by an ensemble of cubic cells (Tao & Nicholson, J. Theoret. Biol. 229: 59 – 68, 2004). The relative diffusivity for a vanishingly small molecule is given by $2/(3 - \alpha)$. Assuming that alpha tracks ECS width then if alpha goes from 0.2 to 0.3, representing a 50% increase, relative diffusivity only goes from 0.71 to 0.74 – a 4% increase

We have modified the text as follows: “According to the restricted diffusion theory, SWCNTs in LB-inoculated mice should display reduced geometric hindrance due to increased ECS width”.

8. There is perhaps no need to introduce the term “extracellular compartment” but rather to go back 50 years to a remarkable volume in the Neuroscience Research Program Bulletin series entitled “The Brain Cell Microenvironment” (Schmitt and Sampson, NRP Bull. 7: 277 – 417, 1969). Sadly, this publication is no longer easily available. This volume was way ahead of its time in focusing on the role of the ECM in brain function and coining the term “brain cell microenvironment”. This term has been used in a number of papers about the ECS. Perhaps “brain extracellular microenvironment” might be even better.

This is an excellent suggestion that we decided to adopt. We have modified all along the text the term “extracellular compartment” by the suggested “brain extracellular microenvironment”, referencing the publication by Schmitt & Sampson in our final remark.

9. What was the length distribution of SWCNTs?

Information on length distribution has been added on page 23, as being centered at around 500nm.

10. Figure 1. Panel e: the volume fraction is based on $n = 3$ mice which seems a bit low for a statistical test. The actual measuring procedure in each mouse presumably takes place over many sections so the n value may be a bit misleading. Identifying the ECS in electron micrographs is sometimes challenging.

We agree with the reviewer. Indeed, identifying the ECS is challenging in EM images. To overcome this limitation, we quantified only images where we were certain that ECS was being segmented (commented in “Methods”). In images with partial freezing artifacts, we draw ROIs around the discernible regions, clearing the outside. Hence, the volume fraction is calculated over the analyzed region and not the entire image. Furthermore, for Cryofixation-EM, since the animals were injected bilaterally and the pathology develops independently in both hemispheres, instead of pooling the SN we have now analyzed them separately. The n is now 6 and expressed as SNs (or ventral midbrains), rather than animals, since the latter is misleading as mentioned by the reviewer.

11. In panel f the dimension scale goes to 0 nm – what is the lower limit of the method? There are some indications in the text that it is around 50 nm. The statement about anisotropy should be considered in the light of earlier comments in this review.

As mentioned earlier, we have modified the scale bars to 50 nm.

REVIEWER 2

1. Many of the findings appear robust and overall the authors present a compelling argument to support their hypothesis. Thus, the data on, for instance, width and diffusion maps of the extracellular space, microglial activation, the degradation of the hyaluronan matrix, cytokine levels and neuroprotective effect of hyaluronidase are compelling. In themselves these are sufficient to make the manuscript of interest and could be highlighted better in the Abstract.

We thank the reviewer for this generous comment. We have modified the abstract (highlighted in yellow) to emphasize the neuroprotective effect of hyaluronan depletion, the activation of microglia, and hinting at ECM manipulation as a disease-modifying strategy. We have also modified the title, to better reflect the general idea that links the matrix (in addition to the pathology) to the alterations observed in ECS parameters.

2. Fig 1e, a trend towards an increase in %ECS in Lewy body treated animals is seen. The analysis appears to only employ $N=3$, if a larger group were used, it is likely that a clearer conclusion could have been drawn. If this were significant, it would be a simple and easily appreciated measure to open the argument for the role of ECS changes in PD.

We appreciate this comment. As commented in point 10 of Reviewer 1, in the cryofixation-EM experiments the animals were injected bilaterally to maximize tissue yield. In these animals, the pathology develops independently in both hemispheres, therefore instead of wrongly pooling the SNs as we did in our original submission, we have now analyzed them separately, in addition to several additional images from unprocessed blocks from the same ventral midbrains. The n is now 6 and expressed as SNs, rather than animals, since the latter is misleading as mentioned by Reviewer 1. We have also analyzed new images taken from previously unprocessed resin blocks to ensure that the trend observed was real. Indeed, the increase in ECS volume fraction from LB-mice is much clearer now. The difference does not achieve statistical significance, although it is very close ($P = 0.064$). It is worth noting that others using this technique *in vivo* have also employed a low n (See Korogod et al., 2015, eLife), possibly due to cryofixation and freeze substitution technique being costly (in terms of time and money). Finally, as we discuss in the manuscript and in the comment below, results on the volume fraction are put into perspective in the light of new methodologies that can reveal local changes instead of parameters that describe an entire region. Hence, we believe that for the general idea of the paper (local changes in the ECS driven by neurodegeneration and alterations in the ECM) the ECS lengths and widths have more weight in the conclusions drawn.

3. In the K-S analyses, e.g. Fig 1f and g, and elsewhere; although several hundred datapoints are entered into the analyses, these only come from very small numbers of animals in each treatment group, for instance $N=3$ in the case of Fig 1f. While the significance suggests that it can be stated that the 3 animals in one group likely had a distribution dissimilar to the 3 animals in the other group, this does not mean that it is so likely that populations of animals treated in this way would be significantly different. The sample is a sample of the two groups of three animals not a sample of the two treatment populations, and thus the statistics are powered for, and assess that comparison.

We have now expressed the results as “mean per SN” (Fig. 1f and 1g, histograms) and analyzed them by t-test, in addition to the KS analyses. Difference of ECS widths and lengths in parkinsonian SN are significant in both cases ($n = 6$). However, we believe that analyzing the data as means does not represent what “local” parameters mean. The whole point of this part of the study is to analyze individual compartments rather than a global “mean” value, and this can only be reproduced by analyzing the continuous population of ECS compartments that represents the wide range of widths and lengths in both groups. Cumulative distributions and KS test show better the skew in the width values of individual ECS compartments (that do not follow a normal distribution and are continuous datasets), and reveal that the difference is more pronounced in the large widths (“pools”, in the terminology used in the manuscript). Mean values and t-tests would mask/dilute this important observation. The ECS volume fraction was calculated different (with a mean) because it is calculated as a global value, not localized, hence it is a mean of several images representing the entire SN. An important point made throughout the manuscript is the importance to add these localized, individual measurements to complement the wealth of data generated over past decades describing “global” parameters of the ECS. A main conclusion of the paper is that even if the volume fractions tends to increase after neurodegeneration, the biggest impact is on how this volume fraction is distributed.

4. The data on the ECM structure in the face of chronic HA reduction are very interesting and could add significantly to the ability to draw conclusions as to cause and effect of ECS dysfunction in neurodegeneration. However, the authors relegate the data showing the effect of this chronic HA reduction on neurodegeneration to the Supplementary materials. This is understandable as the data are inconclusive, but again suffer from low N , $N=4$. If this aspect of the work were properly powered to show a change in neurodegeneration, and a clear effect of treatment were seen, or not, then the conclusions to be drawn would be stronger.

We agree with the reviewer that the low n ($n=4$) is an impediment in the interpretation of the chronic HA modification experiment. However, the low dispersion of the data and the high P value obtained by the statistical test, suggest that the results would not be very different from what we present in Supplementary Fig. 10. The results and their interpretation are, then, substantially different from the experiment where HA was acutely modified and where neuroprotection was observed, and this is an important point that we made clear in the manuscript. Furthermore, we hypothesize that the lack of significant neuroprotection in the 4MU experiment is related to the dual nature of the hyaluronan matrix as diffusional barrier/ECS organizer but also its anti-inflammatory role in HMW configuration. We now present data on Iba1 staining revealing widespread microglial activation after 1 month of 4MU diet even in the absence of synuclein or neurodegeneration (Supplementary Fig. 10c), suggesting that a general activation state of microglia due to long-term hyaluronan depletion might contribute to the lack of neuroprotection. This is indeed a limitation of the 4MU model that have been stated now in the manuscript.

5. The authors use the human-derived Lewy body extracts to induce synucleinopathy. This is a reasonable choice and one of several available models that can produce robust synucleinopathy and degeneration in animals. However, one limitation of the model lies in the ability of the

community to reproduce the findings reported. This limitation stems mainly from the availability of LB extract, either from the same source as used in these studies or from other sources that can be prepared in a way that that produces verifiably-similar protein material. It would thus have been a significant enhancement of the work had they shown replication of some of the key findings with another model of synucleinopathy. Of course, an orthogonal approach to modelling PD, would have further demonstrated the robustness, and likely translatability, to the disease, of the conclusions.

We understand the concerns regarding the reproducibility of the LB model. The protocol for preparation and inoculation of the LB and noLB fractions is easily achievable by any laboratory. The fractions prepared in our teams have been used by many others in the world with similar results regarding seeding, spreading and neurodegeneration. The only limiting factor is the access to human samples. However, in our experience, most laboratories working in proteinopathies or Parkinson's disease have already access to patients' material from brain banks. Although we cannot distribute such precious material, we offer, at the end of the manuscript, to conduct extractions on brain tissues to be provided to us would a colleague want to use similar extracts. Other models of synucleinopathy are significantly more aggressive (and faster, hence their popularity) and/or artefactual, producing side effects related to a several-times fold amount of synuclein in the tissue that affects protein degradation machinery or synaptic communication (exogenous toxic synuclein competing with the endogenous in the formation of the exocytotic pore) in a way very different than the disease. These models use either (i) μg of recombinant alpha-synuclein rather than LB fractions, which normally contain pg of the toxic conformer, or (ii) use viral-mediated overexpression of alpha-synuclein that alters protein processing and triggers massive inflammation. An orthogonal approach such as MPTP or 6-OHDA models are very useful to study dopamine deficiency and circuitry, but do not represent the slow, chronic progression of the pathology and are beyond the scope of this work.

REVIEWER 3

1. Another ECM component should be analyzed alongside hyaluronan to ensure that the effect observed is specific to hyaluronan and not a general effect on the ECM as a whole. This should include staining and quantification of other ECM components after synuclein-induced neurodegeneration and the use of another ECM-degrading enzyme for the Hyase experiments. Currently, the control used is injection of PBS. Injection of an ECM-degrading enzyme that targets another component of the matrix present would show if the neuroprotective effects observed are due to hyaluronan levels, or simply ECM remodeling.

We thank the reviewer for this excellent suggestion. We have analyzed CSPGs in the SN by WFA staining and found no changes neither in the perineuronal nets nor in the interstitial staining found in this region. We have included these results in Fig 3c, which are in accordance to other reports of lack of CSPGS degradation in *in vivo* neurodegeneration in humans (Morawski et al. 2012, Brain Pathol), diminished HA content in mice (Arranz et al., 2014, J Neurosci) or experimental HA degradation in acute slices (Kochlamazashvili et al., 2010, Neuron). Furthermore, we have co-injected ChABC alongside the LBs, effectively degrading the CSPGs locally (Supplementary Fig. 8) and in a similar fashion as in the Hyase experiments, we have analyzed neurodegeneration at 4 months post-injection and neuroinflammation and exogenous alpha-synuclein load at 72 hs post-injection. We found no effect of CSPG degradation on any of these readouts. The compendium of these results is included now in Fig. 5. Changes in the text are highlighted in yellow.

2. The authors state that small MW-HA or hyaluronan breakdown has a neuroprotective effect, as injection of HMW-HA did not show an effect. LMW-HA/HA fragments should also be injected to see if the presence of LMW-HA is sufficient for the neuroprotective effect, or if the endogenous HMW-HA must be degraded.

This is another excellent suggestion. We have co-injected LMW-HA and observed that these fragments alone are sufficient for neuroprotection 4 months after LB inoculation, which in combination with the lack of neuroprotection observed in the ChABC experiment, suggest that HA fragmentation is the key event. There were also not significant trends for decreased alpha-synuclein load and increased microgliosis. These results are included now in Fig. 5. Changes in the text are highlighted in yellow.

3. No neuroprotective effect was shown for 4-MU at 4 months, unlike Hyase. Is there a neuroprotective effect of 4-MU at an earlier time point? That is later lost due to a decrease in Iba1+ microglia levels over time? Levels of microglia activation should be analyzed at an earlier stage (when 4-MU would be expected to start reducing HA levels) to assess if 4-MU alters microglia levels/activation as Hyase did at 72 h. The levels of other ECM components present in the matrix should also be analyzed in these animals to see if that may explain the lack of neuroprotection, as should the levels of the Hyaluronidases.

This is an interesting suggestion. However, due to the slow disease-like progression of the neurodegeneration in the LB mouse model, we cannot assess cell loss at an earlier time point. However, we have analyzed microgliosis in the 4MU model at 72hs after LB injection, in a similar fashion as with the other ECM modification experiments. We found widespread microgliosis after 1 month of 4MU diet (the animals are treated for 4 weeks before the LB injection, to ensure that HA matrix is disrupted at the time of seeding) even in the absence of synuclein or neurodegeneration (Supplementary Fig. 10c). We hypothesize that a general activation state of microglia due to long-term hyaluronan depletion might contribute to the lack of neuroprotection. This is indeed a limitation of the 4MU model that have been stated now in the manuscript. Regarding levels of other ECM components, it has been reported by others (Arranz et al., 2014, J Neurosci) that chronic HA depletion (in this case in Has3^{-/-} mice) does not affect CSPGs, the second most important brain ECM component after HA. Even acute HA degradation by Hyase does not affect the CSPGs levels (Kochlamazashvili et al., 2010, Neuron), although their organization and tridimensional arrangement might be affected (Richter et al. 2018, Curr Opin Struct Biol). This last topic is, however, beyond the scope of this study.

Minor comments

1. The authors state that the width and length of local ECS “pools” is increased, but the total volume of ECS space is not. This is a little contradictory. The total number of pools should also be quantified, as the images in Fig. 1h appear to show there are fewer, larger “pools” present after synuclein-induced neurodegeneration.

We have now quantified the number of pools. As described in the original submission, we had categorized ECS compartments as channels (<100 nm) and pools (>200 nm). We have now included the intermediate category “small pools” (i.e. 100-200 nm) and presented as proportions of total number of ECS compartments. We believe it is an important point of the paper that not only the ECS volume is affected, but especially how this volume is distributed, and the “parts of a whole” representation is suitable to reflect this idea.

2. There are several methodological details that should be included in the results. How long was the imaging of the SWCNTs performed for? How was the mRNA expression analysed? Including this information in the results as well as the methods will enable the reader to follow the experiments outlined more easily.

We agree with the reviewer that including more information in the results section would be ideal. However due to the word limit, we are unable to include all the details about gene expression analysis. However, since the gene expression results in Fig. 3g,h reflect mostly inflammation, we believe this aspect is sufficiently covered in subsequent figures by immunohistochemistry, a technique that not only reports levels of expression, but also location. Regarding nanotube

imaging, we have included a statement about the imaging time of SWCNTs (all changes in text are highlighted in yellow) in the results section.

3. Microglia levels appear to have been quantified by the area of Iba1 staining (Fig 4c), yet Iba1+ soma are used for another quantification (Fig 4f). Iba1+ soma should also be analysed when the level/number of microglia are quantified (Fig 4c, 5j).

We agree with the reviewer. We have now analyzed Iba1+ somas alongside all Iba1 levels. These results are included in Fig. 4c, 5i and Supplementary Fig. 10d.

4. Supplementary videos of SWCNTs in synuclein-induced neurodegeneration brain tissue and the Hyase condition should also be included.

We have now included three additional movies (Supplementary Movies 2, 3 and 4) that show a long-term multiple acquisition, compare noLB vs LB and compare Ctrl diet vs 4MU diet.

REVIEWERS' COMMENTS:

Reviewer #1 (Remarks to the Author):

The authors have addressed all the points I raised about the original manuscript. The paper makes significant new contributions to understanding the extracellular space of the brain on a nanometer scale and the role of hyaluronan in an important pathophysiology of the brain.

Minor Points

The title is ambiguous in that "...diffusion of the brain extracellular space..." may be construed to mean the extracellular space is diffusing. To avoid this problem, "of" could be replaced by "in"; alternatively, the words "properties" or "characteristics" could be inserted after "diffusion".

Lines 293, 494. Perhaps replace "...more thoroughly than..." with "more profoundly than does...".

Line 433. Today, the word "englobe" is mostly restricted to phagocytosis. Perhaps "encompass" would work better.

Charles Nicholson
Professor Emeritus
NYU School of Medicine

Reviewer #2 (Remarks to the Author):

I appreciate the authors' responses to my original comments, the new data that have now been included and changes to the text to aid clarity and interpretation. These responses improve the manuscript and satisfactorily address my original criticisms.

Reviewer #3 (Remarks to the Author):

Soria et al, have performed the requested experiments and analysis and have addressed the concerns stated in the original review, except for the minor point listed below.

3. Microglia levels appear to have been quantified by the area of Iba1 staining (Fig 4c), yet Iba1+ soma are used for another quantification (Fig 4f). Iba1+ soma should also be analysed when the level/number of microglia are quantified (Fig 4c, 5j).

We agree with the reviewer. We have now analyzed Iba1+ somas alongside all Iba1 levels. These results are included in Fig. 4c, 5i and Supplementary Fig. 10d.

I can only find quantification of %Iba1 area in Supplementary Fig. 10d, not Iba1+ cell number as stated. This should be updated to match the quantification present in the main figures.

RESPONSE TO REVIEWERS (2nd round):

REVIEWERS' COMMENTS:

Reviewer #1 (Remarks to the Author):

The authors have addressed all the points I raised about the original manuscript. The paper makes significant new contributions to understanding the extracellular space of the brain on a nanometer scale and the role of hyaluronan in an important pathophysiology of the brain.

Minor Points

The title is ambiguous in that "...diffusion of the brain extracellular space..." may be construed to mean the extracellular space is diffusing. To avoid this problem, "of" could be replaced by "in"; alternatively, the words "properties" or "characteristics" could be inserted after "diffusion".

We appreciate this comment. We agree with the reviewer that the title can be misleading. We chose to replace "of" by "in" rather than to add "properties" for the sake of brevity.

Lines 293, 494. Perhaps replace "...more thoroughly than..." with "more profoundly than does...".

We agree with the reviewer and have replaced the corresponding sentence.

Line 433. Today, the word "englobe" is mostly restricted to phagocytosis. Perhaps "encompass" would work better.

We agree with the reviewer, especially in a manuscript that includes results and discussion about phagocytosis. We have replaced the corresponding sentence.

Charles Nicholson
Professor Emeritus
NYU School of Medicine

Reviewer #2 (Remarks to the Author):

I appreciate the authors' responses to my original comments, the new data that have now been included and changes to the text to aid clarity and interpretation. These responses improve the manuscript and satisfactorily address my original criticisms.

We appreciate the reviewer comments and his/her contribution to manuscript improvement.

Reviewer #3 (Remarks to the Author):

Soria et al, have performed the requested experiments and analysis and have addressed the concerns stated in the original review, except for the minor point listed below.

3. Microglia levels appear to have been quantified by the area of Iba1 staining (Fig 4c), yet Iba1+ soma are used for another quantification (Fig 4f). Iba1+ soma should also be analysed when the level/number of microglia are quantified (Fig 4c, 5j).

We agree with the reviewer. We have now analyzed Iba1+ somas alongside all Iba1 levels. These results are included in Fig. 4c, 5i and Supplementary Fig. 10d.

I can only find quantification of %Iba1 area in Supplementary Fig. 10d, not Iba1+ cell number as stated. This should be updated to match the quantification present in the main figures.

We apologize for the mistake. We have now included the corresponding panel. Now Supplementary Fig. 10d includes both the quantification of area and of cell somas. This is also reflected in the main text.